# Non-consecutive enzyme interactions within TCA cycle supramolecular assembly regulate carbon-nitrogen metabolism

Weronika Jasinska[1,8], Mirco Dindo[2,7,8], Sandra M. C. Cordoba [3], Adrian W. R. Serohijos [4,5], Paola Laurino [2,6] ✉, Yariv Brotman[1] ✉ & Shimon Bershtein [1] ✉

Enzymes of the central metabolism tend to assemble into transient supramolecular complexes. However, the functional significance of the interactions, particularly between enzymes catalyzing non-consecutive reactions, remains unclear. Here, by co-localizing two non-consecutive enzymes of the TCA cycle from *Bacillus subtilis*, malate dehydrogenase (MDH) and isocitrate dehydrogenase (ICD), in phase separated droplets we show that MDH-ICD interaction leads to enzyme agglomeration with a concomitant enhancement of ICD catalytic rate and an apparent sequestration of its reaction product, 2-oxoglutarate. Theory demonstrates that MDH-mediated clustering of ICD molecules explains the observed phenomena. In vivo analyses reveal that MDH overexpression leads to accumulation of 2-oxoglutarate and reduction of fluxes flowing through both the catabolic and anabolic branches of the carbon-nitrogen intersection occupied by 2-oxoglutarate, resulting in impeded ammonium assimilation and reduced biomass production. Our findings suggest that the MDH-ICD interaction is an important coordinator of carbon-nitrogen metabolism.

Metabolism sustains life by generating energy and building blocks through a densely connected network of metabolites and enzymatic reactions organized into linear and branched metabolic pathways. The centrality of metabolism to organismal fitness and survival led to the evolution of complex multi-layered mechanisms that control and regulate the rate of flow of metabolites (metabolic fluxes) through the reaction pathways, as well as the partitioning of theses fluxes in branchpoints[1,2]. Three primary layers of metabolic regulation are traditionally considered: transcriptional control of enzyme expression, post-translational modifications influencing both enzyme abundance and catalytic parameters, and allosteric regulation affecting enzyme activity[1–8]. Another crucial factor affecting flux distribution is the size of metabolite pools[7,9]. In substrate-limited reactions, changes in metabolite concentrations have a direct effect on the reaction rates and thermodynamically favored flux direction. In enzyme-limited reactions, changes in metabolite concentrations can affect flux by competing with other metabolites for enzyme binding sites, due to common structural features shared among many metabolites[9]. Metabolite concentrations are controlled by the kinetics and thermodynamics of enzymatic reactions. Thus, it is generally assumed that cell's capacity to control flux through modulation of metabolite concentrations is

[1]Department of Life Sciences, Ben-Gurion University of the Negev, Beer-Sheva, Israel. [2]Protein Engineering and Evolution Unit, Okinawa Institute of Science and Technology Graduate University, Okinawa, Japan. [3]Max-Planck-Institut fur Molekulare Pflanzenphysiologie, Potsdam-Golm, Germany. [4]Departement de Biochimie, Universite de Montreal, Quebec, Canada. [5]Centre Robert-Cedergren en Bio-informatique et Genomique, Universite de Montreal, Quebec, Canada. [6]Institute for Protein Research, Osaka University, Suita, Japan. [7]Present address: Department of Medicine and Surgery, Section of Physiology and Bio-chemistry, University of Perugia, Perugia, Italy. [8]These authors contributed equally: Weronika Jasinska, Mirco Dindo. ✉e-mail: paola.laurino@oist.jp; brotmany@bgu.ac.il; shimonb@bgu.ac.il

rather limited. However, recent evidence raises the possibility that by forming transient and dynamic supramolecular complexes, metabolic enzymes may enhance flux magnitude and modulate flux partitioning through local enrichment and sequestration of reaction intermediates[10,11].

The existence of local supramolecular enzyme assemblies, known as metabolons, is well-established across various domains of life and pathways, spanning from glycolysis and the TCA cycle to specialized metabolism in plants and fungi[12–20]. Metabolons are thought to control metabolic fluxes through a phenomenon termed 'substrate channeling', whereby reactions intermediates are passed between subsequent active sites, while avoiding equilibration with the bulk solution[10,11,21,22]. This can potentially lead to improved catalytic activity, local substrate enrichment, prevention of degradation of labile compounds, protection from toxic intermediates, changes in reaction equilibrium, and flux shunting between competing pathways[23]. It was proposed that substrate channeling within metabolons may occur via either 'proximity' or 'cluster-mediated' mechanisms. Proximity channeling assumes close positioning of active sites of enzymes catalyzing consecutive reaction, or the presence of patches of charged residues between the active sites of interacting enzymes that electrostatically retain reaction intermediates, thus supporting their efficient transfer between the active sites[18,24,25]. The feasibility of proximity channeling is heavily debated[10,26,27]. An alternative, cluster-mediated channeling mechanism does not require an extreme structural proximity between sequential enzymes[26,28]. Instead, a high local concentration of enzymes is sufficient to achieve substrate channeling by markedly improving the probability of intermediates encountering the next enzyme's active site rather than diffusing away[26].

An intriguing property of some metabolons is that interactions within enzyme assemblies are not only limited to enzymes that are adjacent in a reaction pathway but can likewise occur between *non-adjacent* enzymes belonging to the same metabolic path. A prominent example of such an interaction was detected in both the bacterium *B. subtilis* and the plant *A. thaliana* between malate dehydrogenase (MDH) and isocitrate dehydrogenase (ICD), two non-consecutive TCA cycle enzymes separated from each other by three enzymatic steps[19,29]. Unlike the interaction between consecutive enzymes, the functional significance of non-consecutive enzyme assemblies remains enigmatic, since they cannot participate directly in substrate channeling. It is likely, however, that other enzymes catalyzing intermediary steps between the non-consecutive enzymes, *e.g.*, citrate synthase and cis-aconitase in the TCA cycle, also participate in the supramolecular assembly.

While the evidence for existence of supramolecular enzyme assemblies is ubiquitous, the experimental evidence that links their formation to metabolic flux regulation is often lacking[11]. In this work, we introduce a method using liquid-liquid phase-separated (LLPS) protein droplets to investigate the role of enzyme assemblies in flux regulation. This approach mimics cytoplasmic conditions while enabling direct measurement of enzyme-enzyme interaction effects on flux rate and partitioning. By applying this method, we studied the interaction between the non-consecutive *B. subtilis* enzymes MDH and ICD. We found that the MDH-ICD interaction within protein droplets causes an enhancement of 2-oxoglutarate production rate by ICD with a concomitant increase in 2-oxoglutarate partitioning towards the droplet phase. Theory predicted that the observed phenomena can be explained by the MDH-mediated clustering of ICD molecules. In vivo validation showed that MDH overexpression leads to a 2-oxoglutarate accumulation accompanied by a reduction in fluxes flowing through both the catabolic and anabolic branches of the carbon-nitrogen intersection occupied by 2-oxoglutarate. Our findings suggest that MDH-ICD interaction in *B. subtilis* contributes to the regulation of carbon-nitrogen metabolism.

## Results

### LLPS droplets reveal the functional role of the MDH-ICD interaction

The pairwise interaction between two non-consecutive enzymes of *B. subtilis* TCA cycle was previously identified and validated by co-affinity purifications-mass spectroscopy and bacterial two-hybrid studies[29,30]. MDH catalyzes the conversion of malate to oxaloacetate, while ICD catalyzes the formation of 2-oxoglutarate from isocitrate. To uncover the functional significance of the MDH-ICD interaction under conditions resembling the intracellular environment, we used liquid-liquid phase separated protein droplets. The physicochemical parameters that define the intracellular environment, such as the protein crowding concentration[31–33], cytoplasm viscosity[34,35], and pH values[36,37] are closely matched by the properties of the liquid-liquid phase separated protein droplets[38], indicating that the LLPS system supports enzyme kinetics measurements under the physiologically relevant conditions[39]. As indicated by confocal imaging and activity measurements, ICD and MDH partition within a crowded protein droplet phase (Fig. 1A, Supplementary Fig. 1B). We gradually increased the concentration of MDH within droplets, while keeping the ICD concentration constant (1:1, 10:1, 50:1, 100:1, 250:1, and 500:1 MDH/ICD molar ratios) and measured the ensuing changes in 2-oxoglutarate formation rate. We found that the rate of 2-oxoglutarate formation within droplets was substantially enhanced with the increasing amounts of MDH, starting with ~10% of rate enhancement at 10:1 MDH/ICD ratio and reaching 180% of rate enhancement with MDH/ICD molar ratio at and above 250-fold (Fig. 1B). To validate that the observed effect is rooted in a specific interaction between the two enzymes, we tested whether similar rate enhancements would be achieved upon replacement of *B. subtilis* MDH with its orthologous counterparts. In our previous work, we demonstrated that functional interactions within folate metabolon disappear when the endogenous dihydrofolate reductase-encoding gene is replaced with orthologs[40]. We, therefore, assumed that if the MDH-ICD interaction is indeed functionally important, both enzymes must have been subject to a co-evolutionary process accompanied by the accumulation of reciprocal buffering mutations, making the formation of functional interaction between orthologous MDH and *B. subtilis* ICD less likely. To this end, we expressed and purified MDH molecules from two closely related bacteria, *S. sciuri* and *O. iheyensis*, sharing, respectively, 76% and 82% of amino acid sequence similarity with *B. subtilis* MDH. These enzymes were chosen based on both their high solubility and the similarity of their catalytic activities to that of *B. subtilis* MDH (Supplementary Fig. 2, Supplementary Tables 1, 2 and Methods). Interestingly, ICD orthologs from both *S. sciuri* and *O. iheyensis* also share 83% identity in the amino acid sequence with their counterpart in *B. subtilis*. Replacement of *B. subtilis* MDH with the orthologs in our droplet system produced no detectable enhancement of ICD catalytic activity, even at 1:500 molar excess of orthologous MDHs, suggesting that the observed rate enhancement is rooted in a specific functional interaction between *B. subtilis* MDH and ICD (Fig. 1B and Supplementary Fig. 1A).

In addition to the rate enhancement, we also detected an increase in partitioning of 2-oxoglutarate towards the droplet phase with the increase in MDH abundance. Specifically, a 2-fold increase in the 2-oxoglutarate concentration within droplets loaded with 1:500 molar excess of MDH vs the continuous phase was observed after 40 min of incubation (Fig. 1C). Importantly, in the absence of ICD and MDH, 2-oxoglutarate concentration in both droplet and continuous phases is equivalent, indicating that its diffusion between the phases is unhindered (Supplementary Fig. 1D). Replacement of *B. subtilis* MDH with the orthologs completely abolished the preferential partitioning of 2-oxoglutarate within droplets, reinforcing our earlier conclusion that the observed effects are a result of specific and functional interactions between endogenous ICD and MDH enzymes from *B. subtilis* (Supplementary Fig. 1E, F).

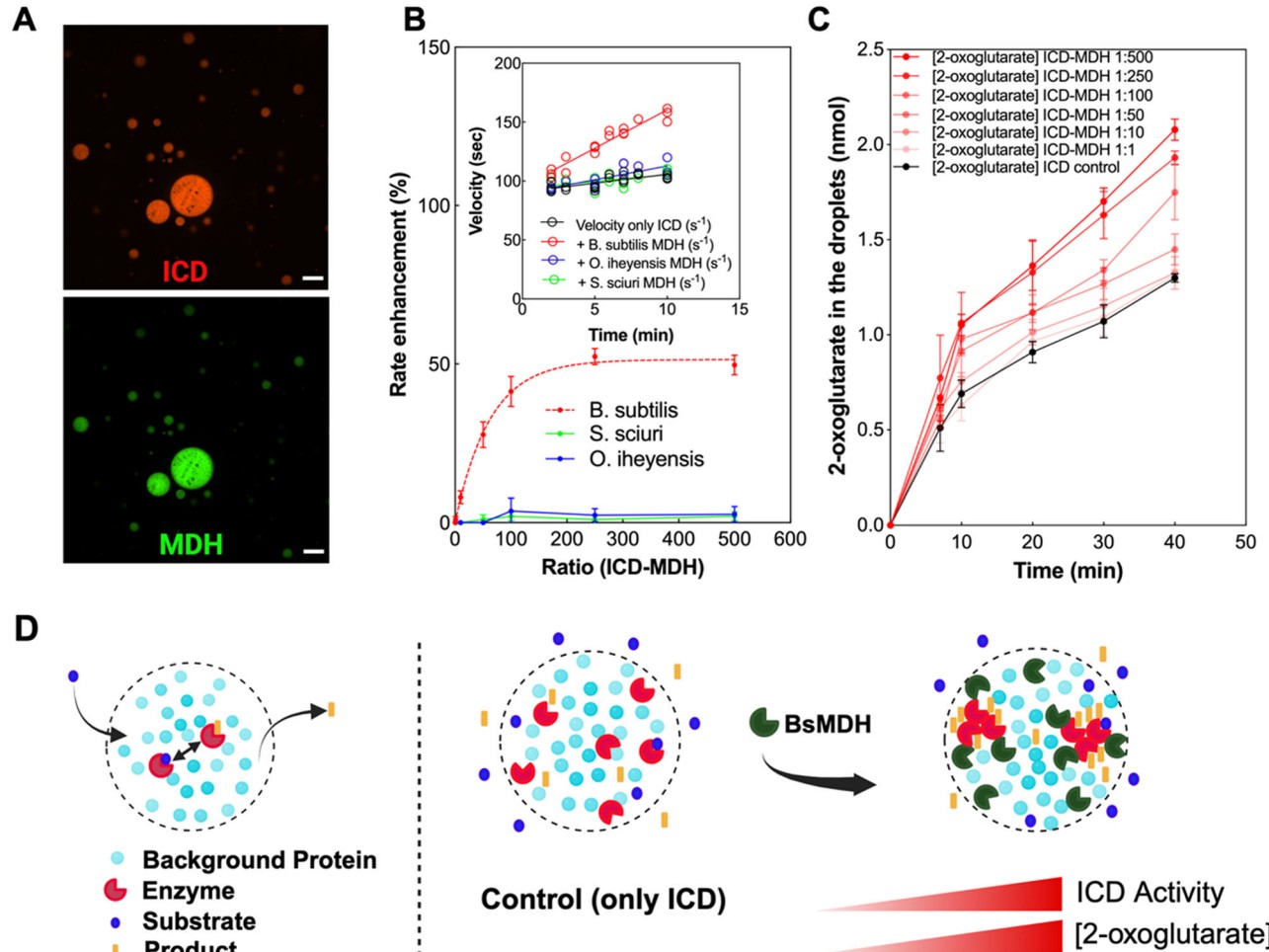

**Fig. 1 | Interaction between *B. subtilis* ICD and MDH within protein droplets causes ICD rate enhancement and product partitioning. A** Confocal microscopy images of ICD and MDH showing the co-localization of the enzymes within droplets. ICD and MDH were labeled with Alexa Fluor 594 (red) and Alexa Fluor 488 (green), respectively. Confocal images were acquired for more than 10 independent experiments with similar results. Scale bars are 20 μm. **B** Rate of 2-oxoglutarate formation (expressed as % of activity) of ICD within droplets is increased in the presence of increasing amounts of *B. subtilis* MDH (1:1, 10:1, 50:1, 100:1, 250:1, and 500:1 MDH/ICD molar ratios), while no effects have been observed in the presence of orthologues MDHs from *O. iheyensis* and *S. sciuri* at the same MDH/ICD ratios. Inset: rate of 2-oxoglutarate formation expressed as ICD velocity (sec$^{-1}$) in the presence of *B. subtilis* and orthologous MDHs. **C** An increase in MDH-ICD ratios within droplets (1:1, 10:1, 50:1, 100:1, 250:1, and 500:1 MDH/ICD molar

ratios) causes an increase in delay of 2-oxoglutarate diffusion into the continuous phase, suggesting that 2-oxoglutarate is sequestered within droplets. Data are represented as mean values ± SD from three independent experiments ($n = 3$). **D** Cartoon of LLPS system that models the reported results. The ICD and MDH enzymes partition within the BSA droplets and the small molecules (substrate, S and product, P) freely diffuse in and out. In the presence of increasing amounts of *B. subtilis* MDH (BsMDH), ICD catalytic rate increases, probably due to MDH-induced clustering of ICD molecules. An increase in ICD activity within clusters causes local accumulation of the reaction product, manifested in the apparent partitioning of 2-oxoglutarate between the droplet and continuous phases. Figure 1D was created with BioRender.com, released under a Creative Commons Attribution-NonCommercial-NoDerivs 4.0 International license. Source data to generate this figure are provided as a Source Data file.

## MDH-mediated ICD clustering explains ICD rate enhancement and product partitioning

What causes the rate enhancement of ICD activity and partitioning of its reaction product within the droplet phase in the presence of MDH? We established that the diffuse state of fluorescently labeled ICD is diminished in the presence of excess amounts of MDH (250:1 MDH/ICD molar ratio) with a concomitant increase in fluorescent puncta, hinting that MDH might trigger or exacerbate clustering of ICD molecules within droplets (Fig. 1D and Supplementary Fig. 3). To evaluate whether MDH-induced ICD clustering might be the cause of the observed phenomena in the LLPS system, we adopted and modified the previously published cluster-mediated substrate channeling theory to model how ICD distribution within droplets affects its activity and product partitioning[26]. In this model, we assume that the entire in vitro system (droplet and continuous phases) can be divided into non-overlapping identical volumes, or "basins", with volume $v_0$ and radius $R_0$ (Fig. 2A). Within each

basin is a droplet of volume $v_2$ and radius $R_2$ containing enzyme $E$ (representing the ICD). The presence of MDH induces enzyme $E$ cluster formation within the droplet, which is represented by volume $v_1$ and radius $R_1$. The number of enzyme $E$ molecules participating in MDH-induced cluster formation is $n_1$, while the number of the remaining $E$ molecules within the droplet but outside the cluster is $n_2$. The total concentration of enzyme $E$ within the droplet is therefore $N_{total} = n_1 + n_2 = n_2(\gamma + 1)$, where the ratio of an excess $\gamma = n_1/n_2$. The cluster formation can now be modeled as an excess in the concentration of the enzyme $E$ in the clusters compared to the droplets:

$$n_1 = N_{total}\left(\frac{\gamma}{\gamma+1}\right); \quad n_2 = N_{total}\left(\frac{1}{\gamma+1}\right) \tag{1}$$

Note that when $\gamma = 0$, there is no cluster, while when $\gamma \to \infty$, all enzymes are in the cluster. In this framework, we can define the

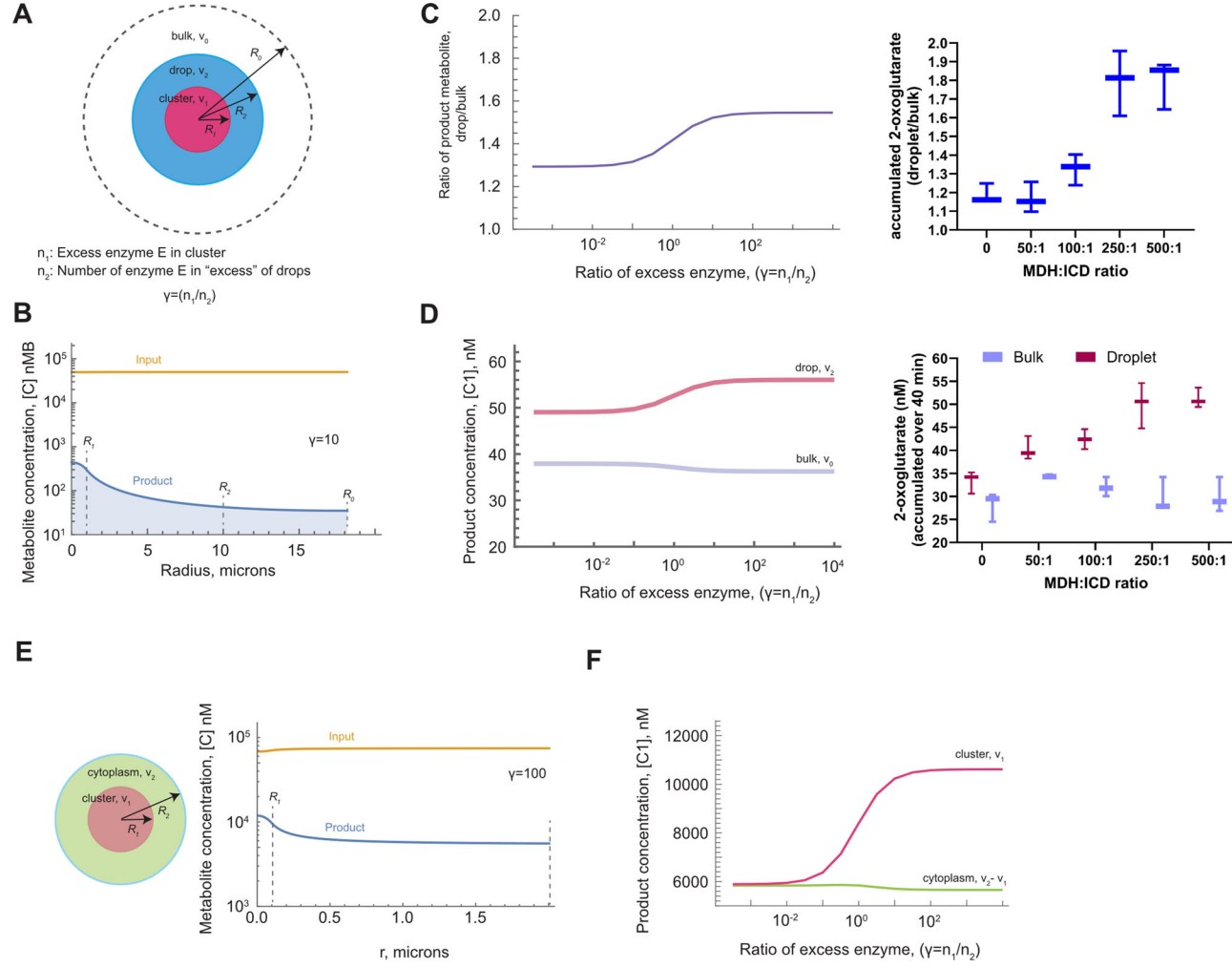

**Fig. 2 | Enzyme clustering leads to an increase in product concentration within clusters. A** Spherically symmetric model of enzyme concentration, whereby enzyme *E* form clusters of radius $R_1$ within droplets of radius $R_2$. The droplet is surrounded by bulk with radius $R_0$. **B** Predicted distribution of product with the different regions of the droplet and basin. The calculations are shown for γ = 10, γ = $n_1/n_2$ (10-fold more enzyme *E* molecules are found within the cluster than in the droplet outside the cluster) **C** *Left panel.* Predicted distribution ratio of total product concentration within droplets vs. bulk as a function of the enrichment of enzyme *E* within the cluster. *Right panel.* Experimental increase in product concentration in droplets vs. bulk as a function of increasing ICD cluster formation mediated by MDH. Horizontal bars represent mean values of three independent experiments (*n* = 3), whiskers indicate spread of individual values from min to max. Ratios in *x* axes correspond to the molar excess of MDH relative to ICD. Zero indicates absence of MDH. Note the agreement between theory and experiment.

**D** *Left panel.* Predicted concentration of a product in bulk and in droplets as a function of increasing cluster formation. *Right panel.* Experimental increase in product concentration in bulk and droplets. Horizontal bars represent mean values of three independent experiments (*n* = 3), whiskers indicate spread of individual values from min to max. Ratios in *x* axes correspond to the molar excess of MDH relative to ICD. Zero indicates absence of MDH *in vivo.* **E, F** Modeling of cluster formation in vivo. **E** Formation of ICD cluster within the cytoplasm (cluster radius is assumed to be $R_1 = 0.1\,\mu m$, cytoplasm size is $R_2 = 2\,\mu m$, see also Supplementary Table 3) causes an increase in product concentration within and in the immediate proximity of the cluster. The calculations are shown for γ = 100 (most of the ICD molecules are found within the cluster). Zero microns correspond to the cluster's center. Input corresponds to the reaction substrate. **F** Predicted substrate concentration (C1) within the cluster and the cytoplasm as a function γ. Source data to generate this figure are provided as a Source Data file and Supplementary Software file.

effective enzyme activity in the droplet, $a_{drop}$, as the product of enzyme catalytic activity $k$ ($k = k_{cat}/K_M$) and enzyme concentration $n_2$:

$$a_{drop} = k\left(\frac{n_2}{v_{drop}}\right) = kN_{total}\left(\frac{1}{v_{drop}}\right)\left(\frac{1}{\gamma+1}\right) \qquad (2)$$

where $v_{drop}$ is the droplet volume. Similarly, the effective enzyme activity inside the cluster, $a_{cluster}$, is

$$a_{cluster} = k\left(\frac{n_2}{v_{drop}} + \frac{n_1}{v_{cluster}}\right) = kN_{total}\left[\frac{1}{v_{drop}}\left(\frac{1}{\gamma+1}\right) + \frac{1}{v_{cluster}}\left(\frac{\gamma}{\gamma+1}\right)\right] \qquad (3)$$

where $v_{cluster}$ is the cluster volume and $n_1$ is the enzyme concentration within the cluster. The cluster, droplets, and basins are assumed to be

spheres and hence the volumes are easily calculable from their radii (Fig. 2A, and Supplementary Notes in Supplementary Information).

Within the cluster and the droplet, the enzymatic reaction *S* (*substrate*)+*E* (*enzyme*)→*P* (*product*) proceeds wherever the enzyme is available, while the metabolites (product and substrate) can diffuse anywhere within the basin. The quantitative modelling of the concentration of metabolites in the volume requires the solution to the coupled reaction-diffusion equation describing the spatial and temporal changes in metabolite concentrations. The substrate $c_0(\vec{r},t)$ is described by:

$$\frac{\partial c_0(\vec{r},t)}{\partial t} = D\nabla^2 c_0(\vec{r},t) - kn(\vec{r})c_0(\vec{r},t) - \alpha_0\left(c_0(\vec{r},t) - c_0^*\right) \qquad (4)$$

where the first term represents the diffusion of the *substrate* into the volume and the second term is the reduction in substrate concentration due to product formation, which is dependent on the enzymatic activity, $k$ and the number of enzyme molecules $n(\vec{r})$. The third term represents the concentration of the substrate which approaches a baseline of a homeostatic concentration $c_0^*$ at a rate $\alpha_0$. Similarly, the reaction-diffusion for the *product* is:

$$\frac{\partial c_1(\vec{r},t)}{\partial t} = D\nabla^2 c_1(\vec{r},t) + kn(\vec{r})c_0(\vec{r},t) - \beta c_1(\vec{r},t) \qquad (5)$$

where the first term is the diffusion of the product into the volume, the second term is the product formation (again dependent on the enzyme activity $k$ and local concentration $n(\vec{r})$). Third term models the decay or incorrect processing of the product, which leads to an effective concentration decay rate $\beta$.

Assuming a steady state and knowing the droplet volume and concentration of $E$ within the volume, Eqs. 4, 5 can be solved exactly to determine the concentration of the product throughout the basin (Supplementary Notes in Supplementary Information). Using experimentally derived parameters (Supplementary Fig. 4, Supplementary Table 3) and appropriate boundary conditions (Supplementary Notes in Supplementary Information), we show the predicted concentrations of the input substrate $c_0$ and the product $c_1$ throughout the basin (Fig. 2B and Supplementary Software). In our case, the substrate is in excess (see Methods). Within the cluster, where there is a local excess of ICD due to the presence of MDH, there is a corresponding increase in product formation. Within the droplet but outside the cluster, where the enzyme is at a lower concentration, the product concentration is correspondingly lower. The presence of the product in the bulk is due solely to its diffusion from the droplet. In other words, the production formation rate within the cluster is higher than the diffusion rate of 2-oxoglutarate outside the cluster. Overall, theory predicts that the concentration of ICD in the clusters creates an effective increase in the product concentration within the cluster. Theory also predicts that an increase in cluster formation will increase the ratio of total product concentration within the droplet vs. bulk (continuous phase) (Fig. 2C and Supplementary Software). This increase will exhibit a quasi-sigmoidal behavior, where in the regime with no cluster formation the concentrations of enzyme $E$ within volumes $R_1$ and $R_2$ are similar. The other limit is reached when all enzyme $E$ molecules are packed within the cluster. However, since the number of enzymes is finite, the effective gain in activity due to the higher enzyme concentration eventually reaches a plateau. Indeed, this behavior agrees with the experimental data, where we vary the extent of ICD cluster formation by increasing the MDH concentration (Fig. 2D and Supplementary Software).

Theory predicts that MDH induced-cluster formation causes both an apparent ICD rate enhancement and partitioning of its reaction product towards the droplet phase. But is this finding relevant to the in vivo conditions? In our LLPS system, the mean droplet size is app. 10 μm, which is roughly an order of magnitude higher than an average size of *B. subtilis* cell (Supplementary Fig. 4). To model ICD clustering under in vivo conditions, we recalculated the predicted concentration of metabolites under the assumption that the droplet size $R_2$ (that emulates the cytoplasm of *B. subtilis* cells) equals 2 μm, whereas the size of MDH-mediated ICD clusters within the cytoplasm is $R_1 = 0.1$ μm (Fig. 2E, Supplementary Table 3, Supplementary Software). We further assume that isocitrate, the ICD substrate, is present in sufficient amounts in vivo, when bacterial culture is propagated in the excess amounts of carbon source. Moreover, the ICD reaction product tends to accumulate within and in the immediate vicinity of the cluster, effectively reducing the availability of the reaction product outside of the cluster. In fact, product accumulation commences with $\gamma = 5$ and reaches a plateau with $\gamma$ approaching 50 (Fig. 2F and Supplementary

Software). Thus, theory predicts that the phenomena caused by MDH-mediated ICD clustering in the LLPS system are also expected to occur in the cell.

## Endogenous but not orthologous MDH overexpression triggers metabolic and fitness effects in B. subtilis

Having predicted that MDH-mediated ICD clustering should cause an apparent rate enhancement of ICD activity and accumulation of its reaction product within (and in close proximity to) the cluster under cellular conditions, we decided to test the contribution of the MDH-ICD interaction to flux regulation in *B. subtilis*. To this end, we over-expressed MDH in *B. subtilis* and followed the ensuing effects on its growth rate and metabolic state. Importantly, as stated above, apart from interacting with ICD, *B. subtilis* MDH also forms interactions with enzymes catalyzing immediate neighboring reactions. Specifically, MDH was found to interact with citrate synthase, a TCA cycle enzyme catalyzing the formation of citrate from oxaloacetate, and with phospho*enol*pyruvate carboxykinase that converts oxaloacetate to phospho*enol*pyruvate – the first step in gluconeogenesis[29,41]. We reasoned that while MDH must not necessarily be active to form a functional interaction with ICD, it certainly must be active to engage in substrate channeling with its consecutive enzymes. Thus, to decouple the functional contribution of the MDH-ICD interaction to metabolic flux regulation from possible substrate channeling effects between MDH and its consecutive enzymes, we generated inactive versions of the endogenous (*B. subtilis*) and orthologous (*O. iheyensis* and *S. sciuri*) MDH enzymes by mutating His at position 180 (according to *B. subtilis* MDH sequence count) at the active site of the enzyme to Ala (Supplementary Data 1 and Methods). This mutagenesis led to the full abolishment of catalytic activity in all MDH variants without affecting either their expression or their solubility (Supplementary Fig. 5A–E). The overexpression of either active or inactive endogenous and orthologous MDH proteins was driven using expression vector pHT254 carrying endogenous *B. subtilis groL* promoter, $P_{Grac100}$, fused with lacI operator to prevent leaky expression before the commencement of exponential growth[42]. The pHT254 vectors were transformed into a laboratory *B. subtilis* strain PY79, and IPTG induction of MDH variants was carried out on the background of the chromosomal MDH expression of the recipient *B. subtilis* strain (Methods). Based on MDH activity measurements in cell lysates (for active variants) and Western Blot analysis, we quantified the overexpression levels of MDH variants and found that upon IPTG induction they were -15-fold higher than the basal level of MDH expressed from the chromosome (Supplementary Fig. 5F–H and Methods).

We began our analysis by testing the effect of MDH over-expression on growth and metabolism of *B. subtilis* cells propagated in the presence of glucose and ammonium. We found that IPTG-induced overexpression of either active or inactive endogenous MDH had a profound effect on both the growth and intracellular metabolic state of *B. subtilis* cells in comparison to uninduced or control (carrying empty pHT254 vector) cells. Although growth rate (doubling time during the exponential growth) was not significantly perturbed, IPTG induction caused a marked decrease in the overall biomass production (or growth yield) after 10 h of growth (Fig. 3A and Supplementary Fig. 6A). Principal component analysis (PCA) of the relative intensities of 657 untargeted metabolic features generated by GC-MS analysis (see Methods) revealed that IPTG induction also produced the greatest amount of variation between *B. subtilis* cells expressing either active or inactive endogenous MDH vs. control cells (66.9% PC1, Supplementary Fig. 7, Supplementary Data 2). Importantly, the intracellular metabolites were extracted at the mid-exponential state ($OD_{600nm} = 0.4$), when the growth effects of overexpression are not yet pronounced (Fig. 3A).

Next, we compared the growth and metabolic effects of IPTG-induced MDH overexpression in *B. subtilis* cells between endogenous

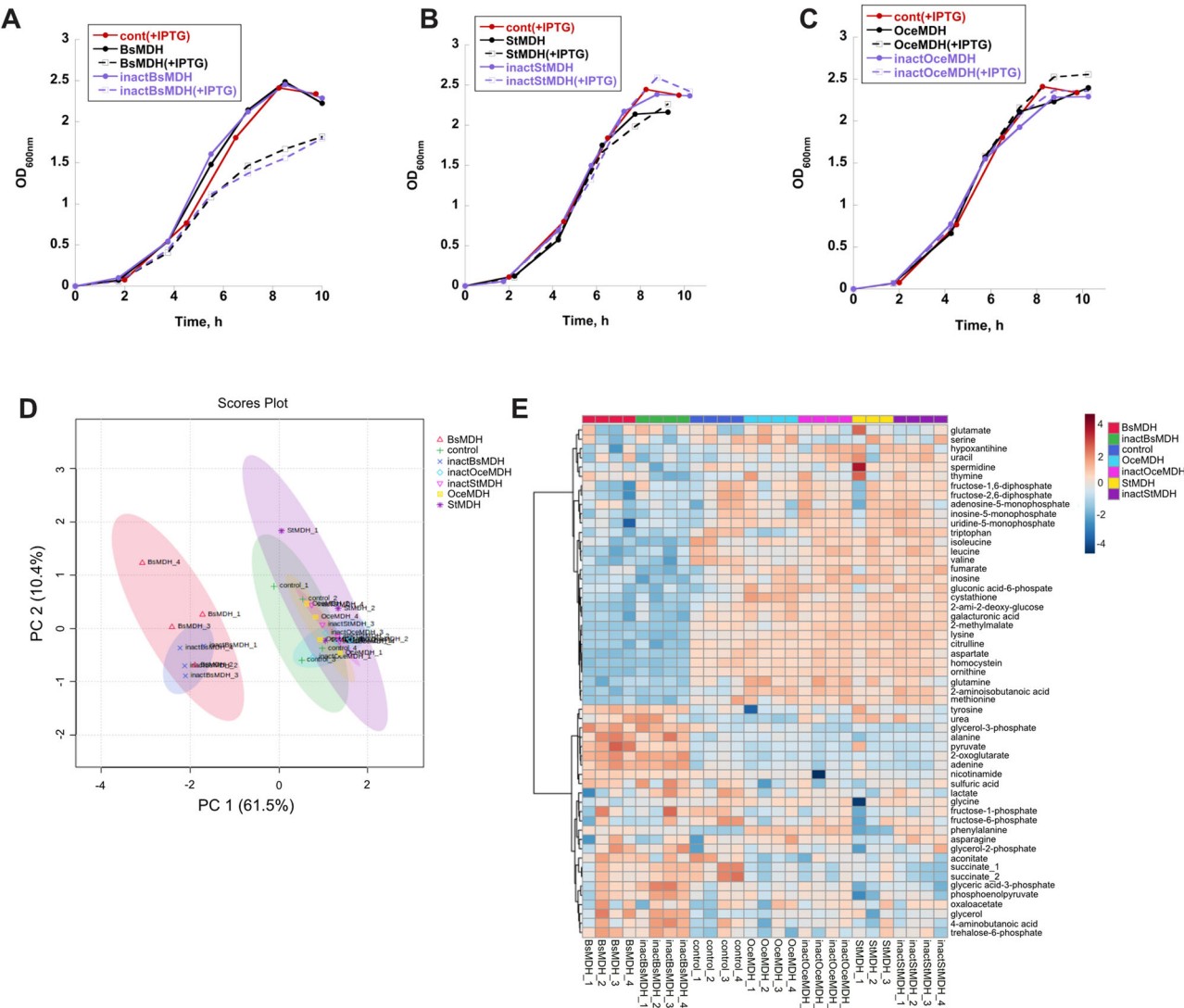

**Fig. 3 | MDH overexpression affects *B. subtilis* growth and biomass production.** **A**–**C** Growth curves of *B. subtilis* cells expressing active (black) and inactive (purple) endogenous (**A**) and orthologous (**B**, **C**) MDHs from pHT254 vector in the presence (solid lines) and absence (dashed lines) of inducer (IPTG). *B. subtilis* cells carrying an empty pHT254 vector are depicted in red. **D** PCA of targeted metabolite intensities identified in four *B. subtilis* replicate populations expressing either active or inactive endogenous and orthologous MDH proteins and control populations. Note the separation on the PC1 axis between *B. subtilis* replicate populations expressing active/inactive endogenous MDH proteins and the rest of the populations. **E** Heat map of -log$_2$ of relative intensities of identified (targeted) metabolites in all *B. subtilis* populations. BsMDH/inactBsMDH, active/inactive *B. subtilis* MDH; StMDH/inactStMDH, active/inactive *S. sciuri* MDH; OceMDH/inactOceMDH, active/ inactive *O. iheyensis* MDH; control, *B. subtilis* carrying empty pHT254 vector. Source data to generate this figure are provided as a Source Data file and Supplementary Data file 3.

and orthologous protein variants, and control cells. Unlike the endogenous MDH, no difference in growth rate and growth yield was detected between IPTG-induced and uninduced *O. iheyensis* and *S. sciuri* MDH orthologs (either active or inactive) or between MDH orthologs and control cells (Fig. 3B, C and Supplementary Fig. 6B, C). Comparison of the relative intensities of 534 untargeted mass features and 55 structurally identified metabolites extracted at the mid-exponential growth phase (OD$_{600nm}$ = 0.4) revealed that while *B. subtilis* cells expressing either active or inactive endogenous MDH proteins are very similar between themselves, they differ substantially from cells expressing orthologous MDH proteins or control cells (PC1 = 60.7%, Fig. 3D, Supplementary Fig. 8, Supplementary Data 3, 4). Furthermore, metabolites extracted from control cells tend to overlap with the orthologous datasets, suggesting that the overexpression of orthologous MDH does not trigger substantial metabolic changes.

Collectively, these findings suggest that, in similarity to our observations in the LLPS system, the growth and metabolic effects are specific to the endogenous MDH overexpression. Furthermore, because both active and inactive endogenous MDH proteins evoke similar effects, we may conclude that they are not triggered by the perturbations in substrate channeling between MDH and its consecutive binding partners.

## Accumulation of 2-oxoglutarate upon MDH overexpression is accompanied by reduced catabolic and anabolic fluxes

What is the nature of the metabolic changes caused by the endogenous MDH overexpression in *B. subtilis* cells? To answer this question, we first repeated the MDH overexpression experiment with active and inactive endogenous MDH variants and control cells, and again found a clear separation in metabolite intensities between MDH-overexpressed and control cells (Supplementary Fig. 9, Supplementary Data 5). We

also found very similar patterns in metabolite ratios between the two experimental datasets (Supplementary Fig. 10). We identified the metabolites responsible for the largest variation in relative intensities between cells expressing endogenous MDH and orthologous MDH/control cells (Fig. 3E). Strikingly, the largest changes upon endogenous MDH overexpression occurred in the intensities of metabolites directly related to the carbon-nitrogen metabolic branchpoint occupied by 2-oxoglutarate. In *B. subtilis* grown on glucose and inorganic nitrogen source (ammonium), glutamate is synthesized de-novo by the activity of glutamate synthase (GOGAT) and glutamine synthetase (GS). GOGAT combines glutamine and 2-oxoglutarate to generate two molecules of glutamate. GS catalyzes glutamine formation from glutamate, ATP, and ammonium – the main metabolic step for utilization of inorganic nitrogen in *B. subtilis*[43] (Fig. 4A). Thus, by providing a carbon skeleton for nitrogen-assimilatory reactions, 2-oxoglutarate occupies a strategic intersection between carbon and nitrogen metabolism[43,44]. Glutamate serves as the major precursors for many nitrogen-containing compounds in the cells, including nucleotides and other amino acids[43]. In *B. subtilis*, glutamate contributes its amino group via 37 known transamination reactions[45]. As a standard free energy of these reactions are close to zero, the directions of metabolic fluxes are believed to be sensitive to the intracellular glutamate levels[9]. We found that while the levels of 2-oxoglutarate have increased by over 3-fold, the levels of metabolites whose de-novo synthesis requires glutamate, including glutamine, aspartate, valine, leucine, isoleucine, ornithine, and citrulline have dropped, on average, by 2–5-fold, hinting that MDH overexpression causes a reduction in metabolic fluxes involving glutamate (Fig. 3E and Supplementary Fig. 11). Comparison of changes in ratios of metabolite pools downstream to branching points revealed a significant increase in the 2-oxoglutarate/glutamate, 2-oxoglutarate/glutamine, and glutamate/glutamine ratios, indicating that the increase in the intracellular 2-oxoglutarate levels upon endogenous MDH overexpression is accompanied by an impeded assimilation of ammonium (Fig. 4A, B). The imbalance in the metabolic pools has also spread to the urea cycle, as manifested in the increase of glutamate/aspartate, glutamate/ornithine, glutamate/arginine, and oxaloacetate/aspartate ratios (Fig. 4C, D). Surprisingly, a significant increase was also found in the ratios between 2-oxoglutarate and downstream TCA cycle intermediates – succinate, fumarate, malate, and oxaloacetate (Fig. 4E, F). Thus, the overexpression of endogenous MDH generated an apparent jam at the carbon-nitrogen branching point: 2-oxoglutarate accumulates within the cells but its utilization in the downstream catabolic (towards succinyl-CoA) and anabolic (towards glutamate/glutamine) paths appears to be impeded. To ensure that these metabolic effects are not triggered by some unrelated pleiotropic effects associated with endogenous MDH overexpression, we replaced glucose, the main carbon source in the growth medium, with glutamine and repeated the metabolomic experiment (see Methods). In *B. subtilis* glutamine is converted directly to glutamate through the activity of glutaminase[46,47], thus potentially alleviating the imbalance in metabolic triggered by MDH overexpression. We reasoned, therefore, that propagation on glutamine should not be accompanied by the metabolic imbalance observed with glucose. If true, we could exclude the pleiotropic effects of MDH overexpression as the main culprit in our observations. After validating that the levels of MDH overexpression in cultures grown on glutamine are comparable to those found in cultures grown on glucose (Supplementary Fig. 12), we propagated the strains expressing either endogenous or orthologous MDHs or carrying an empty expression vector in the presence of glutamine and conducted targeted and untargeted metabolomic analyses. We found that replacement of glucose with glutamine fully removed the metabolic imbalance observed in the presence of glucose, as can be seen in the individual metabolite levels (Supplementary Fig. 13), heat maps and PCA analysis (Supplementary Fig. 14, Supplementary Data 6), and metabolite ratios (Supplementary Fig. 15). Based on these findings

we conclude that the phenomena observed upon endogenous *B. subtilis* MDH overexpression is unlikely to be caused by pleiotropic effects.

To support the conjecture that endogenous MDH overexpression induces metabolic imbalance by causing jamming at the carbon-nitrogen branching point, we measured the changes in rates of catabolic and anabolic fluxes downstream to the branching point occupied by 2-oxoglutarate using the $^{13}C$ tracer approach[48]. Although, in its simplest form, this method does not explicitly allow the determination of fluxes, it does offer a qualitative assessment of the changes in flux rates from the comparison of stable isotope labeling dynamics[48]. For this purpose, we grew *B. subtilis* cells overexpressing endogenous MDH proteins and control cells in the presence of ammonium and 55%/45% of $^{13}C_6$/$^{12}C_6$ glucose and applied GC-MS analysis to follow the metabolite labeling dynamics along the time points collected at a metabolic steady state (Supplementary Fig. 16 and Methods). The interpretation of the labeling dynamics data depends on the changes in metabolite abundances. A delay in the rate of approaching an isotopic steady state of a metabolite whose abundance did not change or dropped, necessarily implies a decrease in flux rate. Conversely, for a metabolite whose abundance increased, such as 2-oxoglutarate, a delay in the labeling dynamics can also be explained by an increase in the size of its metabolic pool[48] (Supplementary Fig. 17). As expected, a significant delay in labeling dynamics was detected for metabolites with decreased abundances in both catabolic (succinate, fumarate) (Fig. 5A–D) and anabolic (ornithine, valine) (Fig. 5E–H) branches, indicating that MDH overexpression reduces metabolic rates in the metabolic paths downstream to 2-oxoglutarate.

## Metabolic shifts induced by MDH overexpression mimic nitrogen starvation

When glucose and ammonium are abundant, the increased cellular demand for amino acids and nucleotides is satisfied by routing the biosynthetic carbon, in particular oxaloacetate and 2-oxoglutarate, from the TCA cycle towards anabolic reactions. Whereas oxaloacetate contributes carbon scaffold for aspartate synthesis (Fig. 4C), 2-oxoglutarate is siphoned towards glutamate/glutamine production (Fig. 4A). 2-oxoglutarate levels tend to fluctuate in response to carbon-nitrogen availability, making it an intracellular signaling molecule recruited by the cell to coordinate carbon-nitrogen metabolism[43,44,49,50]. When both glucose and nitrogen are in excess, 2-oxoglutarate is used extensively for glutamate/glutamine production, and its intracellular levels are low. The drain of 2-oxoglutarate from the TCA cycle is balanced in *B. subtilis* by the anaplerotic activity of pyruvate carboxylase that converts pyruvate into oxaloacetate[41]. When glucose is abundant, but nitrogen is limited, the drain of 2-oxoglutarate and, correspondingly, the anaplerotic flux drop, while the intracellular concentration of 2-oxoglutarate raise[41,44,49]. The observed accumulation of 2-oxoglutarate and impeded utilization of ammonium detected upon MDH overexpression are therefore reminiscent of the intracellular metabolic state under the conditions of nitrogen starvation. To further explore the resemblance of the metabolic changes incurred by MDH overexpression to nitrogen starvation, we tested the effect of overexpression on anaplerosis. We found a significant increase in the pyruvate levels, and an increase in a pyruvate/oxaloacetate ratio, suggesting a perturbation in the anaplerotic carbon flux towards the TCA cycle (Fig. 4G, H and Fig. 3E). We also found a significant increase in the phospho*enol*pyruvate/fructose-1,6-diphosphate ratio (Fig. 4G, H). Since the free energy difference of the glycolytic/gluconeogenic reactions connecting these two metabolites are close to zero, the increase in the ratio between the metabolites of low and upper glycolysis suggests a reduction in the glycolytic flux[9].

Collectively, these findings demonstrate that, despite the presence of an excess amount of ammonium in the growth medium, MDH overexpression triggers a metabolic state that emulates growth under nitrogen limitation: an increase in 2-oxoglutarate levels, a reduction in

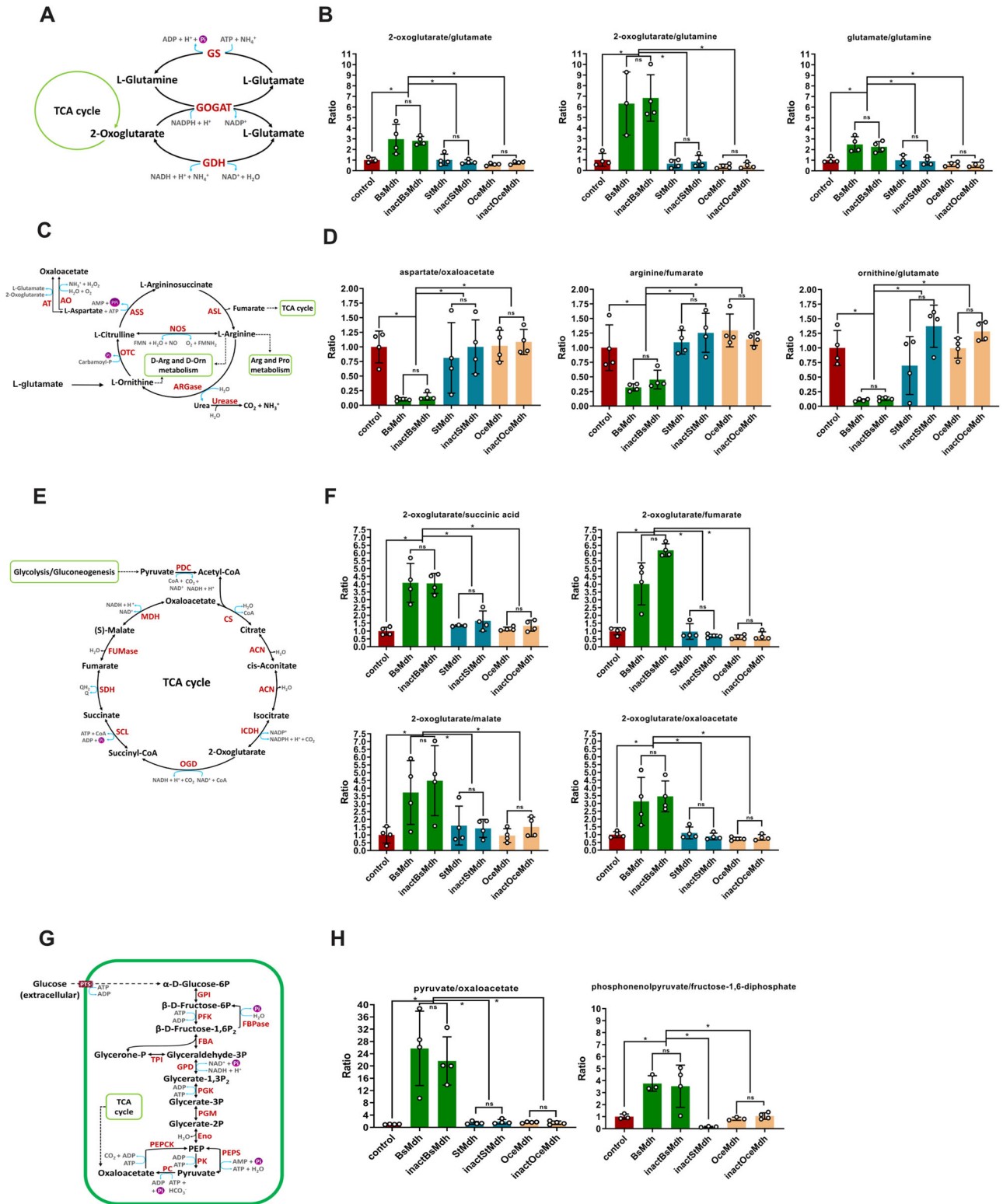

**Fig. 4 | MDH overexpression affects the metabolite ratios. A** Scheme of glutamate/glutamine synthesizing/degrading pathways in *B. subtilis*. **B** Changes in metabolic ratios between metabolites related to scheme A. **C** Scheme of urea cycle biosynthetic reactions in *B. subtilis*. **D** Changes in metabolic ratios between metabolites related to scheme C. **E** Scheme of TCA cycle biosynthetic reactions in *B. subtilis*. **F** Changes in metabolic ratios between metabolites related to scheme E. **G** Scheme of glycolysis/gluconeogenesis in *B. subtilis*. **H** Changes in metabolic ratios between metabolites related to scheme. **G** For clarity, ratios between relative intensities of metabolites in control cells were set to 1, and the corresponding ratios in strains expressing MDH proteins were normalized accordingly. In the metabolic schemes, enzymes are shown in red, products/substrates in black, cofactors in blue. **B**, **D**, **F**, **H** Histograms represent mean values ± SD from three to four independent biological replicates (*n* = 3–4). Individual values are shown as empty circles. Asterisks indicate significant differences (*p* = 0.029, two-tailed Mann Whitney test). ns, non-significant. BsMDH/inactBsMDH, active/inactive *B. subtilis* MDH; StMDH/inactStMDH, active/inactive *S. sciuri* MDH; OceMDH/inactOceMDH, active/inactive *O. iheyensis* MDH; control, *B. subtilis* carrying empty pHT254 vector. Source data to generate this figure are provided as a Source Data file.

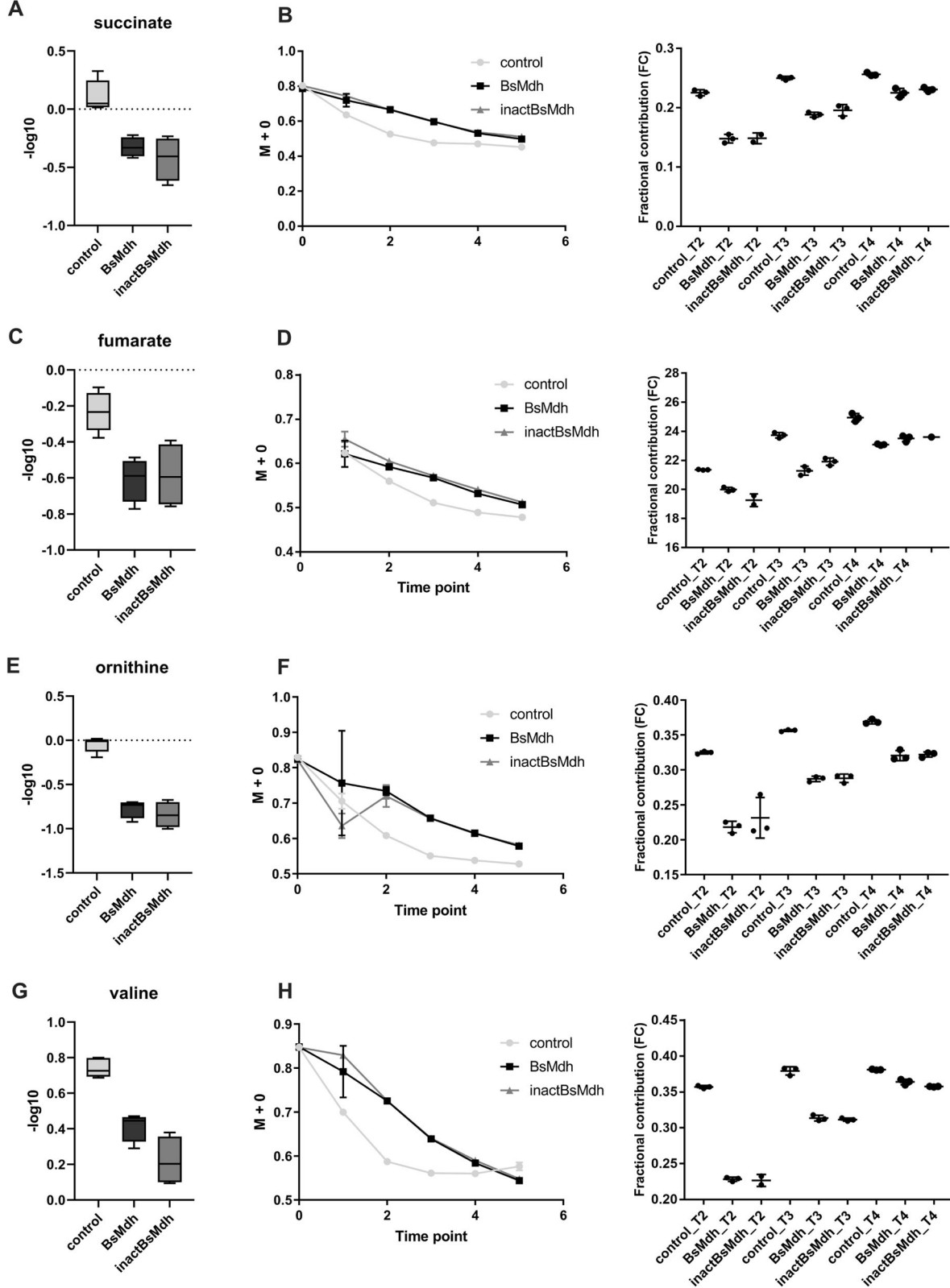

**Fig. 5 | ¹³C tracer analysis reveals changes in metabolic fluxes. A**, **C**, **E**, **G** -log10 of relative intensities of succinate, fumarate, ornithine and valine in *B. subtilis* strains expressing active and inactive endogenous MDH (BsMDH and inactBsMDH, respectively) from IPTG-induced pHT254 vector or carrying an empty vector (control). Boxes represent tenth to ninetieth percentile. Horizontal bars represent mean values from three independent measurements (*n* = 3). Whiskers represent the spread of individual values from min to max. **B**, **D**, **F**, **H** *Left panels*. Drop in fractional abundance of unlabeled isotopologues (M + 0) due to labeling with heavy

isotope of glucose (¹³C). Time points 2–4 correspond to metabolic steady state. Data are represented as mean values ± SD from three independent experiments (*n* = 3). *Right panel*. Fractional contribution of the fully ¹³C-labeled metabolite. Time points T2-T4 correspond to metabolic steady state. Horizontal bars represent mean values for three independent experiments (*n* = 3). Vertical bars indicate ± SD. Individual values are shown as dots. Source data to generate this figure are provided as a Source Data file.

ammonium assimilation, and a perturbation in carbon flux distribution at the phospho*enol*pyruvate-pyruvate-oxaloacetate intersection.

## 2-oxoglutarate sequestration alone does not explain the metabolic shifts induced by MDH overexpression

To interpret the intracellular metabolic changes caused by MDH overexpression, we consider two, potentially complementary, mechanisms. The first mechanism assumes a sequestration of 2-oxoglutarate through its retention within MDH-ICD clusters. This mechanism is supported by the LLPS measurements and by the enzyme clustering model described above. The second mechanism assumes a stoichiometric disruption of the TCA cycle supramolecular complex. Indeed, apart from MDH, ICD forms interactions with two downstream consecutive enzymes catalyzing catabolic and anabolic pathways of the metabolic carbon-nitrogen intersection occupied by 2-oxoglutarate. Specifically, ICD interacts with E2 subunit of 2-oxoglutarate dehydrogenase complex catalyzing the formation of succinyl-CoA (the catabolic branch), and with a large subunit of GOGAT catalyzing glutamate formation (the anabolic branch)[29]. It is possible, therefore, that MDH overexpression causes partial ICD sequestration, reducing or fully severing its interaction with the downstream enzymes[51]. To test the feasibility of these mechanisms, we supplemented MDH-overexpressing *B. subtilis* cell growing on ammonium and glucose with dimethyl-2-oxoglutarate, which, after freely diffusing through the membrane, is converted into 2-oxoglutarate in the cytoplasm[49]. We reasoned that if the metabolic shifts induced by MDH-overexpression are caused predominantly by the retention of de-novo synthesized 2-oxoglutarate within the MDH-ICD complex, 2-oxoglutarate supplied through the growth medium should fully restore the metabolic imbalance. Conversely, if the observed metabolic effects are caused, at least partly, by the stoichiometric disruption of the TCA cycle supramolecular complex, addition of 2-oxoglutarate to the growth medium will not be sufficient to reverse the metabolic changes. As expected, addition of excess amounts of dimethyl-2-oxoglutarate to the growth medium fully equalized the intracellular 2-oxoglutarate levels between control and MDH-overexpressing cells, indicating that dimethyl-2-oxoglutarate was converted to 2-oxoglutarate in the cytoplasm (Supplementary Fig. 18A). Glutamine levels were also equalized between conditions (Supplementary Fig. 18). However, it did not lead to the equilibration of the metabolic states between the strains, as manifested by the PCA of the relative intensities of targeted metabolites (Supplementary Fig. 19, Supplementary Data 7). Although 2-oxoglutarate/succinate and 2-oxoglutarate/glutamine ratios have decreased by app. 2-3-folds, they did not reach the levels of control cells, suggesting that fluxes through the catabolic and anabolic branches were not fully restored (Supplementary Fig. 20A, B). In fact, 2-oxoglutarate/glutamate ratio increased by 3-fold upon addition of dimethyl-2-oxoglutarate (Supplementary Fig. 18 and Supplementary Fig. 20C). In addition, we detected a dramatic reversal of the metabolic ratios in oxaloacetate-pyruvate and glutamate-glutamine junctions, manifested in drop in pyruvate levels and decrease in pyruvate/oxaloacetate and glutamate/glutamine ratios in the MDH-overexpressing cells (Supplementary Fig. 18 and Supplementary Fig. 20D, E). Thus, despite the increase in 2-oxoglutarate availability, the metabolic impact of MDH overexpression was not restored, suggesting that 2-oxoglutarate availability alone cannot explain the metabolic changes induced by MDH overexpression.

## Discussion

To better understand the functional significance of the interaction between two non-adjacent enzymes of the *B. subtilis* TCA cycle metabolon, MDH and ICD, we used liquid-liquid phase separated (LLPS) protein droplets. We found that with the increase in MDH/ICD ratio in the droplet phase, 2-oxoglutarate production rate within droplets was

enhanced, while 2-oxoglutarate partitioning towards the droplet phase became more pronounced. Crucially, no such effects were observed when *B. subtilis* MDH was replaced with orthologous MDHs from closely related bacteria, suggesting that in *B. subtilis*, MDH and ICD are subject to a selection pressure to maintain functional interaction. We determined that under the assumptions of enzyme clustering model, the observed phenomena can be explained by an MDH-mediated clustering of ICD molecules within protein droplets. Theory predicts that high local concentration of ICD molecules within clusters causes an enhancement of 2-oxoglutarate production rate and, concomitantly, an emergence of the effective gradient of 2-oxoglutarate concentration that peaks at the center of the cluster and monotonically decreases towards the bulk (continuous phase). Although the diffusion rate of 2-oxoglutarate between droplets and bulk is unhindered, the apparent partitioning of the product towards the droplet phase is observed because the rate of product formation within ICD clusters exceeds the rate of its diffusion outside the droplets.

Theory also predicts that when the radii of clusters, droplets, and bulk are re-scaled to match the environment of a prokaryotic cell, the phenomena observed in the LLPS system (i.e., ICD rate enhancement and 2-oxoglutarate concentration gradient) should still hold. Thus, we attempted to evaluate the role of MDH-ICD interaction within the complexity of the cellular milieu. In similarity to what was observed in the LLPS system, in vivo overexpression of MDH caused an accumulation of 2-oxoglutarate. In fact, the metabolic effect spread far beyond 2-oxoglutarate, and involved multiple metabolic pathways in which glutamate serves a precursor, including amino acid synthesis (glutamine, aspartate, valine, leucine, isoleucine), the TCA cycle biochemical reactions downstream of 2-oxoglutarate, and the urea cycle. We also detected a perturbation at the phospho*enol*pyruvate-pyruvate-oxaloacetate metabolic crossroad, which is key to distribution of carbon fluxes between glycolysis, gluconeogenesis, and anaplerosis. 2-oxoglutarate accumulation was accompanied by a reduction in catabolic and anabolic fluxes downstream to 2-oxoglutarate. We considered two non-mutually exclusive mechanistic explanations to the observed metabolic state. The first explanation assumes that 2-oxoglutarate accumulation in the immediate vicinity of MDH-induced ICD clusters results in a reduced availability of this metabolite in the downstream reaction, causing all the observed effects, including an impeded nitrogen assimilation and cellular growth defects. The second explanation assumes that ICD clustering severs (at least partially) its interaction with the consecutive downstream enzymes, effectively breaking the TCA cycle metabolon integrity due to MDH overexpression-induced stoichiometric imbalance. To distinguish between these two possibilities, we supplemented the growth medium with the external source of 2-oxoglutarate and found that the metabolic state was not reversed. Based on these findings, we conclude that the reduction in 2-oxoglutarate availability alone is not sufficient to explain the observed phenomena, and another mechanism, such as the disruption of the TCA cycle metabolon, must be at play.

Although the demonstration of liquid-liquid phase separation of the TCA cycle in vivo is challenging and is a topic of future research, our findings demonstrate that in vitro LLPS system constitutes a readily available system to explore the functionality of specific enzyme-enzyme interactions within metabolons under controlled conditions that capture the reality of crowded intracellular environment. The accompanying theoretical model in combination with in vivo metabolic analysis further facilitate the interpretation of the obtained in vitro measurements and provide the opportunity to delineate in a comprehensive manner functional interactions between consecutive and non-consecutive enzymes within any cellular metabolon and link them to the accompanying regulatory mechanisms of metabolism. Using this approach, we currently attempt to map the functional interactions between all the components of the TCA cycle

metabolon in *B. subtilis* and uncover the structural elements that have evolved to support these interactions. Importantly, metabolic regulation via metabolon formation is prevalent in both primary and secondary metabolism[10,11]. The proposed experimental and theoretical frame, therefore, opens an exciting venue for harnessing the finesse of metabolic regulation at metabolon level for the purposes of metabolic engineering.

## Methods

### Materials
Polyethylene glycol (PEG) 4000 Da (A16151) was purchased from Alpha-Aesar. Alpha-ketoglutarate Assay Kit (MAK054), Bovine serum albumin (BSA) (A7638), Potassium phospate dibasic trihydrate (60349), Potassium phosphate monobasic (P0662) 3-(Trimethoxysilyl) propylmethacrylate (440159), 2-Hydroxy-4′-(2-hydroxyyethoxy)-2-methylpropiophenone (410896), Poly(ethylene glycol) diacrylate (PEGDA) 700 Da, were purchased from Sigma-Aldrich. Alexa Fluor 594, 488 Microscale Protein Labeling Kit (A30008 and A30006), HisPur Ni-NTA resin (88221) and polypropylene empty columns (R64050) were purchased from Thermo Fischer Scientific. DL-Isocitric acid (19608-51) and NADP (23340-36) were purchased from E-Nacalai Tesque.

### Protein expression and purification
*E. coli* BL21 cells transformed with vectors encoding ICD or MDHs were grown in 1 l of Luria broth at 37 °C to an absorbance at 600 nm of 0.4–0.6. Expression was induced after 6 h with 1 mM IPTG for 15 h at 25 °C. Cells were harvested and resuspended in 20 mM sodium phosphate buffer pH 7.4, containing 0.5 M NaCl and 40 mM imidazole. Lysozyme was added to a final concentration of 0.2 µg/ml and the culture was incubated for 15 min at room temperature. After a freeze-thaw the suspension was centrifuged at 30,000 g for 30 min at 4 °C and the lysate was loaded in pre-packed column containing Ni-NTA resin previously equilibrated with 20 mM sodium phosphate buffer pH 7.4 containing 0.5 M NaCl and 20 mM imidazole. The protein was eluted using the same buffer containing 500 mM imidazole. The proteins have been extensively washed using 100 mM KP buffer, pH 7.0, in Amicon Ultra 10 concentrators (Amicon). The purity of the protein was confirmed by analysis of the SDS-PAGE gel.

### Protein labeling
Isocitrate dehydrogenase (ICD from *B. subtilis*) and malate dehydrogenase (MDHs from *B. subtilis*, *S. sciuri* and *O. iheyensis*) were tagged using Alexa Fluor 488 and 594 Microscale Protein Labeling Kit, using the manufacturer's protocol but repeating the dialysis step three times to remove the free dye in the protein solution. Tagged proteins were further purified using membrane dialysis to reduce the free probe in solution. To determine the protein concentration of ICD from *B. subtilis* we used $\varepsilon_{280nm} = 55,475\,M^{-1}\,cm^{-1}$ (https://web.expasy.org/protparam/) while for MDH the molecular weight used for protein quantification were the following (*B. subtilis* = 33,643 Da, *S. sciuri* = 34,377 Da and *O. iheyensis* MDH = 34,592 Da) (https://web.expasy.org/protparam/).

### Droplet formation
A polyethylene glycol stock solution at 600 mg/mL was prepared mixing 6 grams PEG 4000 Da with the appropriate amount of Milli-Q water to 10 grams. The pH of the solution was measured to be around 7 with a pH meter. A bovine serum albumin stock solution with a target concentration of 5-7 mM (332 mg/mL) was prepared mixing BSA with Milli-Q water. The actual concentration was confirmed by measuring the absorption intensity at the 280 nm peak with a UV-vis spectrophotometer (UV-1900 UV-Vis Spectrophotometer, Shimadzu), assuming $\varepsilon_{280nm} = 43,824\,M^{-1}\,cm^{-1}$ (https://web.expasy.org/protparam/). The pH of the solution was measured to be around 7. In our working conditions, we mixed the components with a final target concentration of 150 mg/mL of PEG, 140 mg/mL BSA, 200 mM KCl, and 100 mM KP pH 7.0. In detail, the reagents have been added and mixed following a specific order: first we added the appropriate volume of buffer KP 0.2 M pH 7.0 and small molecules (enzymes, cofactors, salts and substrates at the appropriate concentration); after mixing, we added BSA at the final concentration of 447 µM. After mixing, as last component of the system, PEG was slowly added (23% w/w final) and the solution has been carefully mixed, and the composition of the droplets was checked qualitatively at the confocal microscope. If required, the supernatant phase was isolated centrifuging the droplet suspension for 30 min at 16,900 g at 25 °C and extracting the supernatant phase with a pipette (without withdrawing the droplet phase).

### Imaging studies and ICD fluorescence analysis
All fluorescence images were acquired and analyzed using a Spinning Disk Confocal (Nikon, Andor CSU) microscope with ×63 oil immersion lens. Confocal images were acquired for at least 5–10 independent experiments with similar results. The qualitative analysis of the ICD intensity fluorescence obtained in presence of partitioned endogenous and orthologues MDHs has been performed by using the Spinning Disk Confocal microscope (Nikon, Andor CSU). In detail, we have prepared a fluorescence intensity calibration curve using three different concentrations of labeled ICD (using Alexa Fluor 594) which correspond to 0.5 µM, 2 µM and 5 µM inside the droplets which linearly fit in the intensity plot (Fig. S2 panel A). At this point, we have analyzed the fluorescence intensity of ICD (2–2.5 µM) alone and partitioned with 400–500 µM MDHs (ratio 1:200, as final concentrations within the droplets) (MDHs from *B. subtilis*, *O. iheyensis* and *S. sciuri* have been used). This analysis was performed by measuring the fluorescence intensity of a fixed ROI inside the droplets. At least 5–10 droplets per condition (different MDH) have been analyzed and the images collected were analyzed using ImageJ.

### Determination of 2-oxoglutarate
The determination of the 2-oxoglutarate in the droplet and supernatant and in the supernatant alone was performed as follows. 1 mL of droplets resuspension containing 100–120 nM ICD (in the droplets) has been prepared as reported in the previous section (Droplet formation) and assayed using a final concentration of 5 mM NADP and 10 mM DL-isocitric acid. Briefly, 1 mL of droplet resuspension containing ICD 100–120 nM has been added to 1 mL of supernatant alone (droplet were removed by centrifugation at 16900 g for 1 h at 25 °C) containing 10 mM NADP and 20 mM DL-isocitric acid. Then, at each time point 300 µL of reaction mix was withdrawn and centrifuged for 1 min at 16900 g (obtaining the droplet phase and the supernatant phase). 10 µL of droplet phase and 10 µL of supernatant phase were diluted in 40 µL of KP 100 mM pH 7.0 (50 µL final volume) in two separated microcentrifuge tubes and immediately stopped by heating the samples at 90–100 °C for 2 min using a water bath and stored in ice for 10–20 min. Then, after 30 min of centrifugation at 16,900 g at 4 °C (to remove the denatured BSA and enzymes), the 2-oxoglutarate concentration in the droplet and supernatant phase was measured using the alpha-Ketoglutarate Assay Kit (colorimetric mode), and the samples were analyzed using a Multiskan Sky High microplate reader (Thermo Fisher Scientific). In detail, in a 96-well plate 2.5 to 5 µL of the stopped and centrifuged reaction mix was added to the specific alpha-ketoglutarate assay buffer to a final volume of 50 µL. Then 50 µL of Master Reaction Mix (alpha-ketoglutarate Converting Enzyme + alpha-ketoglutarate Development Enzyme Mix + Fluorescent Peroxidase Substrate) was added to the 96-well plate and the samples were mixed by shaking for 15–30 s. The reaction was incubated for 30 min (protecting the plate from the light) at 25 °C monitoring the absorbance at 570 nm every 5 min. 2-oxoglutarate provided by the Kit was used to

make the calibration curve in buffer and supernatant, as shown in Supplementary Fig. 1C (0–10 nmoles 2-oxoglutarate is the range used for the calibration curve) as well as to test the partitioning of the 2-oxoglutarate in the droplet and supernatant phase as reported in Supplementary Fig. 1D. The values of absorbance of the samples have been converted in nmoles using the calibration curve and converted in Velocity using the concentration of enzyme and the time of the assay. To evaluate the partitioning of ICD, we measured the activity of the enzyme in the supernatant phase that remained after droplets were removed by centrifugation. In detail, 600 µL of droplets and supernatant containing 100–120 nM ICD was split (300 µL each) into two separated tubes. At this point, one tube has been centrifuged for 30 min at 16,900 $g$, 25 °C to completely remove the droplet phase. After transferring the supernatant to a fresh microcentrifuge tube, another centrifugation cycle was applied with the same settings as before to be sure all the droplets were removed from the supernatant phase. Then, 250 µL of supernatant containing final concentrations of 5 mM NADPH and 10 mM DL-Isocitric acid were added to 250 µL of droplets and supernatant, and supernatant alone. Finally, we stopped, assayed, and analyzed the samples as described above.

## Similarity network of MDH from Bacterial Kingdom

Data for the MDH sequence similarity network (Fig. S2) were obtained from the website EFI – ENZYME SIMILARITY TOOL (https://efi.igb.illinois.edu/efi-est/) by uploading the amino acids sequence of malate dehydrogenase of *Bacillus subtilis* PY79 strain. The data were visualized with Cytoscape (https://cytoscape.org/).

## Bacterial strains and plasmids

Original and mutated *mdh* genes used in the study are summarized in Supplementary Data 1. *B. subtilis mdh* gene was amplified by PCR from the chromosomal DNA of *B. subtilis* PY79. Nucleotide sequences of orthologous *mdhs* were collected from the KEGG database (https://www.genome.jp/), synthesized by GENEWIZ© and cloned into i) pHT254 *B. subtilis* expression vector (MoBiTec Molecular Biotechnology), and ii) pET24a *E. coli* expression vector (Merck-Novagen) using Gibson assembly. The inactive MDH mutants were generated by site-directed mutagenesis by replacing predicted catalytic residues (https://www.uniprot.org) to Ala (see Supplementary Data 1). All MDH proteins were tagged with Hisx6t at the C-terminal end. Mdh genes cloned into pHT254 plasmid were transformed into *B. subtilis* PY79. Mdh genes cloned into pET24a plasmid were transformed into *E. coli* Bl21(DE3) strain.

## MDH expression and purification

MDH proteins were overexpressed in *E. coli* BL21 (DE3) in LB supplemented with 100 µg/ml kanamycin. Overnight liquid cultures started from a single colony were diluted 1:100 and grew at 37 °C until $OD_{600}$ reached 0.5. IPTG was then added to the culture to a final concentration of 1 mM and the induction continued at 30 °C overnight. Cells were collected by centrifugation at 4500 × $g$. Lysis was performed for 30 min on ice with addition of lysozyme (Sigma), final concentration 1 mg/ml, and 500 U benzonase (Sigma). The next step was sonication (Q125 Sonicator, Qsonika Sonicators). The lysate, filtered by a syringe-driven filter unit (PVDF 0.45 µm, Millex®), was then purified by Ni-NTA using a His-TRAP FF 5-ml column (GE Healthcare®). Collected elutions were dialyzed into 10 mM HEPES, pH 7.5, 150 mM NaCl, and 3 mM EDTA. This step was followed by size-exclusion chromatography using Superdex 200 16/600 column (GE Healthcare). The proteins were concentrated by centrifugal filters (Amicon Ultra 10 K, Merck Millipore Ltd.). Protein concentrations were determined by a Pierce™ BCA Protein Assay Kit (Thermo Fischer Scientific) and their purity validated by sodium dodecyl sulfate - polyacrylamide gel electrophoresis (SDS-PAGE) followed by Western blot (WesternBreeze Chromogenic Immunodetection System, Thermo Fisher Scientific)

using mouse antibodies against His-tag (Qiagen, Cat. No. 34660) at 1/5 dilution.

## Purified MDH activity assay

Determination of MDH activity was based on spectroscopic assays (30). The activity of all active and inactive purified MDH proteins was estimated by following NADH to NAD+ conversion in the presence of oxaloacetate by following the reaction progress at 340 nm. Specifically, a range of oxaloacetate (Sigma) concentrations (7.5–250 µM), 150 µM NADH (Sigma), 5 nM MDH in 0.1 M sodium phosphate buffer, pH 7.4 were used. Inactive mutants showed less than 0.1% of their wild-type counterparts. The catalytic parameters, $k_{cat}$ and $K_M$, were calculated by fitting the data using non-linear regression to Michaelis-Menten equation, $v_O = (k_{cat} \cdot [E] \cdot [S])/(K_M + [S])$, where $v_O$ is initial velocity, $[E]$ is MDH concentration, $[S]$ is NADH concentration.

## MDH activity assay in cell lysates

*B. subtilis* PY79 strains transformed with pHT254 carrying endogenous (*B. subtilis*), or orthologous (*O. iheyensis* and *S. sciuri*) *mdh*, or an empty vector (control strain) were propagated in liquid LB cultures supplemented with 5 µg/ml chloramphenicol at 37 °C (tarted from a single colony). After reaching $OD_{600} = 0.4$, the cultures were diluted 1:100 in 50 ml SMM supplemented with 5 µg/ml chloramphenicol and 1 mM IPTG and grew at 37 °C until the $OD_{600}$ reached 0.3–0.45. Pellets were collected by centrifugation at 4 °C for 15 min at 4000 × $g$ and lysed for 30 min on ice in the presence of 1 mg/ml lysozyme (Sigma) and 500 U benzonase (Sigma). The next step was sonication (Q125 Sonicator, Qsonika Sonicators), centrifugation at 4 °C for 15 min at 4000 × $g$ and filtration by a syringe-driven filter unit (PVDF 0.45 µm, Millerpore). Protein concentration in the filtrates was measured by a Pierce™ BCA Protein Assay Kit (Thermo Fischer Scientific) and unified across the samples. MDH activity in the lysates was measured the presence of saturating amounts of oxaloacetate (3 mM), NADH (1 mM), and 5 ml of a cell lysate to ensure maximal velocity conditions. The reaction velocity was estimated by following NADH to NAD+ conversion. The MDH concentration, $[E]$, in the cell lysates was calculated assuming maximal velocity ($V_{max}$), $[E] = V_{max}/k_{cat}$. The $k_{cat}$ values were estimated from purified MDH activity assays (see above).

## Propagation of *B. subtilis* cultures

B. subtilis strains were grown in Spizizen minimal medium (SMM) (40) supplemented with 0.4% glucose (Sigma) or 0.4% glucose and 4 mM dimethyl-2-oxoglutarate or 8 mM glutamine (Sigma). Growth protocol for metabolite profiling of *B. subtilis* strains (40) was adjusted for our purpose: liquid culture in LB was started at 37 °C from a single colony from an overnight LB plate. After the $OD_{600}$ reached 0.4, the culture was diluted 1;100 in 50 ml SMM supplemented with 5 µg/ml chloramphenicol and a corresponding carbon source and propagated with shaking at 37 °C until the $OD_{600}$ reached 0.3–0.45.

## MDH expression and aggregation in B. subtilis strains

Cells were harvested at $OD_{600}$ of 0.3–0.45, then centrifuged at 4 °C for 15 min at 4000 × $g$. The supernatant was discarded, and pellets were diluted in 1× BugBuster® (10× Protein Extraction Reagent, Novagen® Merck), 50 mM $NaH_2PO_4$ pH 8, 1 mM lysozyme (Sigma), and 500 U benzonase (Sigma). After 30 min incubation on ice and 20 min on the roller, samples were centrifuged for 20 min and supernatants were separated from pellets. Next, the pellets were washed first with 1× BugBuster® (10× Protein Extraction Reagent Novagen® Merck), then twice with water and once with 50 mM $NaH_2PO_4$ (pH 8). At this step the total protein concentration within the supernatant and pellet fraction for each sample was measured by QPRO-BCA Kit Standard (Cyanagen). The obtained results were used to normalize the samples. The last step was to dissolve pellets in 5× SDS loading dye and boil them in 99 °C for 15 min, whereas 5× SDS loading dye was diluted by the sample and

water 1:5 and boiled for 10 min in 99 °C. Samples were then loaded on a Mini-PROTEAN TGX Gel 4–20% (BIORAD). After the run, gels were stained in Instant Blue (Expedeon) and after 1 h washed with water.

## Metabolite extraction and GC–MS analysis

The protocol for metabolite extraction was adjusted for our purpose (41). Cells (between three to four replicates per each strain/condition) were harvested at $OD_{600}$ of 0.3–0.45, then centrifuged at 4 °C for 15 min at $4000 \times g$. The supernatant was discarded, and the pellet was resuspended in 1 ml of ice-cold phosphate-buffered saline and centrifuged again in the same conditions. Afterwards, ice-cold 80% ultrapure methanol was added to the pellet, mixed well by vertexing, put for 10 s in liquid nitrogen, and incubated for 30 min at −20 °C. To remove cell debris, samples were centrifuged at a maximum speed at 4 °C for 30 min. The supernatant was collected into a fresh Eppendorf tube and by using a SpeedVac vacuum concentrator (Thermo Fischer Scientific) the liquid was evaporated. Dry samples were derivatized (42) and the metabolites were measured by GS–TOF–MS. Data extraction for untargeted metabolites was performed by extraction of mass features by XCMS™ (https://xcmsonline.scripps.edu/) and clustering to unique metabolite using an R script. Targeted analysis was performed by Xcalibur™ Software (Thermo Fisher Scientific) and supported by TargetSearch. Normalization was done based on OD measurements of harvested cells. MetaboAnalyst 5.0 (https://www.metaboanalyst.ca/) and GraphPad Prism 10 (Dotmatics) was used for statistical analysis and presentation of the data like PCAs, heatmaps, box plots and metabolic ratios. Two-tailed Mann Whitney test was used to define the significant difference between the ratios and boxplots, where $p < 0.05$ is considered as significant. The extracted metabolomics data are provided in Supplementary Data files 1–7. The raw metabolomics data have been deposited in the MetaboLights database under accession code under accession code MTBLS10225.

## $^{13}$C-labeled glucose tracer analysis

$^{13}$C-tracer analysis was performed on *B. subtilis* strains transformed with empty pHT254 plasmid (control), and pHT254 plasmids carrying active and inactive endogenous mdh genes. The strains were grown as described above, with the exception that SMM Medium was supplemented with 55%/45% of $^{13}C_6/^{12}C_6$ glucose. Samples (3 replicates per each strain) were collected at 5 time points: T0 – before the dilution in SMM, T1 – after 3 h, T2 – after 4 h, T3 – after 5 h, T4 – after 6 h, and T5 – after 7 h. The protocol from sample collection to data extraction was the same as in the section *Metabolite extraction and GC- MS analysis*. Extracted data contained not only abundances of main feature masses but also abundances of their isotopologues. Based on this, the mass distribution vector (MDV) was calculated for each metabolite according to ref. 43. MDVs are used to describe the fractional abundance of each isotopologue, where the maximum is equal 1 due to the normalization to the sum of all possible isotopologues. The fractional contribution (FC) of the $^{13}$C-labeled glucose to a metabolite's carbon was calculated from MDV, using equation:

$$FC = \frac{\sum_{i=0}^{n} i \cdot s(i)}{n}$$

where $i$ denotes isotopologue, $s(i)$ is the relative fraction of the isotopologues, and $n$ is the number of carbon atoms[48]. The metabolic steady state was determined by $i$) demonstrating steady rates glucose consumptions; and $ii$) demonstrating steady levels of metabolites. Glucose uptake rate was calculated as described in ref. 37. Three independent experiments were conducted in which sugar concentration in the medium and cell dry weight were measured after 3, 4, 5, and 6 h. The sugar consumption rate was estimated by fitting the data to $DS = (−q/m) \cdot DC$, where $S$ is the glucose concentration in the medium (g/L), C is the cell dry weight in grams (gCDW), and m is the

growth rate (1/h) of the cultures. The glucose concentration in the medium was determined by colorimetric assay with anthrone reagent (Sigma) in a 96 well plate. Anthrone was dissolved in sulfuric acid and immediately added to the samples and standards. Then, the plate was wrapped in aluminum foil and placed in the oven for 30 min at 80 °C. The plate was cooled down and absorbance was measured at 620 nm.

## Reporting summary

Further information on research design is available in the Nature Portfolio Reporting Summary linked to this article.

## Data availability

The data generated in this study and required to reproduce analyses and figures are provided with this paper as a Source data file and Supplementary Data files 1–7. The raw metabolomics data have been deposited in the MetaboLights database under accession code MTBLS10225. Source data are provided with this paper.

## Code availability

The code used in this study is provided as a Supplementary Software file (Mathematica 13.3.1., Wolfram).

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

## Acknowledgements

We are grateful to Alisdair Fernie, Eugene Shakhnovich, and Barak Rotblat for the useful discussions and comments on the manuscript. We thank Gudrun Walter for their help in acquisition of GC-MS data. S.B. was supported by the Israel Science Foundation personal research grant 593/21. P.L. was supported by the Okinawa Institute of Science and Technology Graduate University (OIST) with subsidy funding from the Cabinet Office, Government of Japan. M.D. thanks the financial support from Japan Society for the Promotion of Science (JSPS) for the Kakenhi Early Career Scientist N.22K15065.

## Author contributions

Conceptualization: S.B., Y.B., P.L. Methodology: S.B., Y.B., P.L., A.W.R.S., M.D., S.M.C.C., W.J. Theory: A.W.R.S. Investigation: M.D., W.J., S.M.C.C. Visualization: M.D., W.J. Supervision: S.B., Y.B., P.L. Writing—original draft: S.B. Writing—review & editing: M.D., P.L.

## Competing interests

The authors declare no competing interests.
