## [Peer Review File · Nature Communications]

Reviewers' Comments:

Reviewer #1:

Remarks to the Author:

In the manuscript entitled "Non-consecutive enzyme interactions within TCA cycle supramolecular assembly regulate carbon-nitrogen metabolism", Jasinska et al. described the evidence of metabolic functions of the association of non-consecutive enzymes, MDH and ICD, in *Bacillus subtilis*. The in vitro experiments using a liquid-liquid phase-separation model system showed inclusion of ICD and its catalytic product, 2-oxoglutarate, in the LLPS protein droplets and activation of ICD activity when excess MDH was in the system regardless of its catalytic activity. The simulation demonstrated that it could happen in the scale of bacterial cells. The metabolite profiling and ¹³C tracer experiments of the *B. subtilis* cells overexpressing catabolically active and inactive MDH both showed the accumulation of 2-oxoglutarate and reduced flux through the glutamine biosynthesis and the TCA cycle.

This research addresses the long-standing question of the functions of non-consecutive enzymes and provide the clear in vitro evidence for the activation of ICD in LLPS protein droplets formed with MDH. The in vivo overexpression study also showed clear differences in cellular metabolic status by overexpressing active and inactive MDH. I found this study exciting as it provide a novel insight into the metabolic regulation mechanism, and this study should have significant impacts to the research field. The manuscript is well written and easy to follow, although I am not sure if I fully understand the modeling part due to my background knowledge on biophysics.

I have several concerns.

1. I wonder if the use of lac promoter inducible plasmid is suitable for the metabolic analysis. I'm not sure in *B. subtilis*, but in *E. coli*, IPTG induction of lac promoter controlled protein expression direct most cellular resources to synthesize proteins and the cellular metabolism is not in a physiological state. Is the lac promoter driven expression a reasonable system to test metabolic effects of the expressed protein in *B. subtilis*? I understand that the MDH from other organisms serves as the control, but the use of lac-IPTG system is still need to be justified.
2. Is there any evidence for MDH-ICD LLPS formation in *B. subtilis* cells?
3. When MDH overexpression reduces the flux to nitrogen fixation, how do the cells produce a large amount of MDH protein? *B. subtilis* also secrete proteins into culture media, making amino acid consumption even larger.
4. The empty vector cell is not a suitable control for the tracer experiment since it does not express any protein. The large flux allocation to protein synthesis should affect metabolic network regardless of the expressed protein.
5. The conclusion only summarizes the findings. I want to know more about the impact of the findings, still missing information, and future directions.

Reviewer #2:

Remarks to the Author:

In this manuscript, Jasinska et al. study the relevance of protein-protein interactions between enzymes that catalyze non-consecutive steps in a metabolic pathway. It is generally assumed that enzymes of metabolic pathways assemble to so-called metabolons. Typically, these interactions are weak and difficult to reconstitute in vitro. It is also assumed that the interaction between enzymes that catalyze consecutive reactions of the pathway serves to shuttle metabolic intermediates directly from one enzyme to the next. However, the published interaction between the malate dehydrogenase (MDH) and isocitrate dehydrogenase (ICD) in the TCA cycle in the bacterium *B. subtilis* doesn't meet this criterium as the enzymes catalyze distant steps in the pathway thus questioning the relevance of the interaction. A previous study by Bartholomae et al. (2014) has already shown that the complex formation results in enhanced ICD activity. This result is confirmed in the current study using a different experimental approach as compared to the original paper. Using a combination of experimental and mathematical analysis the authors show that the interaction results in ICD clustering which is responsible for the enhanced enzyme activity. In

addition, the authors show that overexpression of MDH results in accumulation of 2-oxoglutarate which is unexpectedly accompanied by reduced fluxes in associated reactions (such as amino acid transamination). Unfortunately, the reason for the latter observation could not be clarified.

Reviewer #3:

Remarks to the Author:

This study aims at characterizing how the interactions between non-consecutive enzymes of a metabolic pathway can impact the pathway's functionality. Taking MDH and ICD as a test case, both from *Bacillus subtilis* where the enzymes are part of a larger metabolon, the authors combine in vitro LLPS experiments, modelling, in vivo metabolomics and ¹³C tracer analysis to do so. Results suggest that the interactions between MDH and ICD play an important role at the interface between C and N metabolism in *B. subtilis*.

The manuscript is clearly written. The authors attempted to implement appropriate controls and to deconvolute the observed effects (e.g. use of inactive enzyme variants to avoid direct metabolic channelling effects...), which is appreciable. The conclusions are interesting and overall well supported by data. However, I have a few comments listed hereafter that should be considered to improve the quality of the draft.

Lines 92-99 : It is intriguing that MDH and ICD can interact... but in vivo, it is likely that other enzymes catalysing intermediary steps (e.g. CS and CAN, mentioned on line 245 for CS) also from part of the supramolecular assembly. This should be mentioned here, as direct substrate channelling may well represent one of the metabolon properties when considering the whole picture.

Line 122: The claim that "liquid-liquid phase separated protein droplets" resemble the intracellular environment seems a bit bold to me, or at least it needs some justification (Is it because of the high molecular crowding? Of other biochemical properties? Re-creating the cellular environment in vitro represents a long-lasting challenge, see for example: <https://doi.org/10.1016/j.pisc.2014.02.011>).

Figure 1 (and corresponding text): The percentage of rate enhancement when adding *B. subtilis* MDH saturates at high MDH:ICD ratios (somewhere around 200). Are such high ratios really physiologically relevant? What happens when much lower ratios (e.g. 1:1 or 5:1... or even 1:5, any data available?) are utilized, which probably mimics better what would happen in the intracellular environment?

Panel 1D, right: I got a bit confused because it looks like the quantity of ICD within the droplet increases after MDH addition (schematically, 5 ICDs on the left but 7 on the right). However, the quantity of ICD is not supposed to change, right? This panel should be modified.

Line 138: Could you also indicate the percentage of identity between ICDs from these different organisms? Is there any information available as to whether the MDH-ICD interactions also exist within these organisms?

Lines 158-160 and Fig.S3: Signal intensity within the droplet diminishes in presence of *B. subtilis* MDH and it seems that some "fluorescent puncta" concomitantly appear (Fig.S3B lower panel). Based on confocal microscopy observations, are these puncta indeed only visible in presence of *B. subtilis* MDH? Are these puncta also visible at lower MDH:ICD ratios? This is important because it could represent the only direct experimental evidence of ICD clustering...

Line 213-215: If I understand correctly, the production rate within the cluster is higher than the diffusion rate of 2-oxoglutarate outside the cluster, explaining why the local concentration

increases? This should be stated more clearly.

Figure 2: Is Fig. 2D showing actual experimental data? Please add "measured product concentration" so the reader can directly see these are empirical results, not results from modelling. Same thing about panels 2D and 2F (I assume these are experimental results but it is not very clear based on the legend... also, what do the error bars represent?).

Lines 224-225: You hypothesize that substrate is present in a large excess. Is it really compatible with what happens intracellularly, i.e. where only limited amounts of isocitrate are available? Please add a sentence to discuss about the last point, because it is important concerning the in vivo relevance of the proposed model.

Line 240: Can you really state based on your model that "clustering should cause rate enhancement of ICD activity"? Clustering does not seem to have any effect on the ICD (specific) activity, it rather leads to a local increase in ICD concentration and therefore in 2-oxoglutarate formation. I would be careful with phrasing.

Fig. S5: Panels A) and B) seem to be inverted in the figure.

Fig.3E and comparison with Fig. S9: If I understand correctly, these heat maps refer to results of targeted metabolomic experiments performed on separate occasions. There seems to be some discrepancies within the results when looking into details: in Fig.3E 2-oxoglutarate (and pyruvate) appear relatively more abundant in the BsMDH (active or inactive) expressing strains than in the controls, but the tendency is not clear in Fig.S9. How reproducible are these metabolomic experiments (besides the PCA showing a comparable separation of samples) and can we really conclude that there is a 2-oxoglutarate accumulation when over-expressing BsMDH?

Lines 393-394: Overexpressing BsMDH may have other pleiotropic effects that could only be assessed using a systems biology approach (e.g. transcriptomics, quantitative proteomics...). It is possible that the highly accumulated BsMDH triggers regulatory responses and that the abundance/activity of other key central metabolism enzymes consequently changes. The impacts would go far beyond a "stoichiometric disruption of the TCA cycle supramolecular complex" and could also explain the observations made in presence of deimethyl-2-oxoglutarate.

We thank the editors and the reviewers for their careful consideration of our manuscript. Our point-by-point responses to the reviewers' comments are below; their original comments are in italics. All changes introduced to the revised text are highlighted in red.

Reviewer #1

1. I wonder if the use of lac promoter inducible plasmid is suitable for the metabolic analysis. I'm not sure in B. subtilis, but in E. coli, IPTG induction of lac promoter controlled protein expression direct most cellular resources to synthesize proteins and the cellular metabolism is not in a physiological state. Is the lac promoter driven expression a reasonable system to test metabolic effects of the expressed protein in B. subtilis? I understand that the MDH from other organisms serves as the control, but the use of lac-IPTG system is still need to be justified.

B. subtilis has a rather small gene overexpression toolbox. Initially, we tested the effect of $P_{\text{hyperspank}}$ promoter on MDH expression from *amyE* chromosomal location¹. However, this system did not produce any measurable increase in MDH levels (Fig. 1 below). A different expression system that eventually was used in our work is based on an *endogenous B. subtilis groL* promoter, P_{Grac100} , fused with *lacI* operator to prevent leaky expression before the commencement of the exponential growth². Unlike the strong IPTG-inducible promoters used for protein overexpression in *E. coli*, e.g., P_{tac} or T7, that can indeed cause a massive overexpression of a recombinant protein (up to 30% of the entire soluble proteome), the levels of overexpression of MDH proteins in *B. subtilis* in our system are rather modest (Fig. 1 below), and are no more than 10-15 fold increase relative to the endogenous baseline, as estimated from Western Blot analysis. Given that similar levels of overexpression of orthologous MDH proteins *i)* do not produce any measurable effect on bacterial growth (Fig. 3A, main text; Fig. S6, Supplementary Materials), and *2)* do not cause an alteration in the metabolic state of cells relatively to cells carrying an empty vector (Fig. 3D,E, main text; Fig. S8, Supplementary Materials), we can conclude that the effect of MDH overexpression does not have a substantial effect on the physiological state of *B. subtilis* cells.

Fig. 1. Comparison of MDH overexpression levels obtained with $P_{\text{hyperspank}}$ and P_{Grac100} -based expression systems in *B. subtilis* using Coomassie-stained protein gel. Left panel. The first 4 lanes (up to the dashed red line) depict cell lysates extracted from WT *B. subtilis* strain. The lanes after the red dashed line show cell lysates extracted from *B. subtilis* cells carrying IPTG-inducible $P_{\text{hyperspank}}$ promoter placed upstream to *mdh* coding gene within *amyE* locus of the chromosome¹. The location of the anticipated MDH overexpression is marked with a red arrow. Numbers on top of the gel indicate IPTG levels in mM. Right panel. MDH overexpression from IPTG-

inducible pHT254 plasmid carrying *mdh* gene located downstream to P_{Grac100} promoter². The visible MDH overexpression is marked with a red arrow. Numbers on top of the gel indicate IPTG levels in mM.

2. *Is there any evidence for MDH-ICD LLPS formation in B. subtilis cells?*

Apart from the demonstration that MDH and ICD participate in a direct physical interaction *in vivo*³, no evidence for their involvement in LLPS formation *in vivo* was produced. There is an inherent difficulty, however, in demonstrating a direct interaction between MDH and ICD *within* LLPS *in vivo*, simply because other enzymes of TCA cycle metabolon could in principle mediate the LLPS formation. This is precisely the reason why using an *in vitro* LLPS system, in which the identity and abundance of the partners participating in cluster formation can be directly controlled, provides a direct advantage in studying metabolons.

3. *When MDH overexpression reduces the flux to nitrogen fixation, how do the cells produce a large amount of MDH protein? B. subtilis also secrete proteins into culture media, making amino acid consumption even larger.*

The drop in biomass production upon endogenous MDH overexpression becomes apparent rather late in the growth cycle, upon entering the stationary phase (around OD₆₀₀=1) (Fig. 3A, main text). However, the measurements of the MDH abundance and the metabolic effects of overexpression are done at the early exponential phase at OD₆₀₀=0.4, when no growth effects are detected (Fig. 3A, main text). These findings indicate that the interference with the flux to nitrogen fixation does not have an *immediate* effect on growth, and therefore there is no immediate major effect on protein synthesis. Rather, the effect of the metabolic imbalance on growth triggered by MDH overexpression is slow and cumulative and becomes apparent only when the growth cycle is entering the stationary phase, *i.e.*, when the cellular need for protein synthesis subsides. This conclusion is also supported by the fact that the overexpression of endogenous MDH affects only the final biomass levels (yield) and has no effect on the growth rate measured during the mid-exponential phase, *i.e.*, when the cellular demand for protein synthesis is the highest (Fig. S6, Supplementary Materials).

4. *The empty vector cell is not a suitable control for the tracer experiment since it does not express any protein. The large flux allocation to protein synthesis should affect metabolic network regardless of the expressed protein.*

The empty vector would indeed be an unsuitable control under conditions in which the observed metabolic effects are caused by resource allocation due to protein overexpression. However, a range of measurements ensure that this is *not* the case. Specifically, we demonstrated that an independent overexpression of two orthologous MDH proteins at the levels identical to those of the endogenous MDH does not cause any measurable effects on bacterial growth or on the intracellular metabolic levels. In fact, both the growth effects and

the metabolic status of the strains expressing orthologous MDH proteins are indistinguishable from those measured in the strain carrying an empty vector, which proves that resource allocation due to overexpression is not a substantial contributing factor (Fig. 3, main text; Fig. S8, Supplementary Materials). Overall, this is not surprising, given the rather modest levels of MDH overexpression (Fig. S5B, Supplementary Materials). To further validate that this conclusion is robust also for the tracer experiment, we calculated the relative abundance ratios for a set of representative metabolites in strains carrying an empty vector or expressing endogenous MDH protein under the conditions of tracer experiment and compared them with the corresponding ratios obtained from strains carrying an empty vector or expressing endogenous and orthologous MDH proteins and propagated under regular growth conditions. As can be seen in Fig. 2 below, the metabolite ratios are highly robust to the growth conditions and follow identical patterns, thus re-enforcing the conclusion that resource allocation due to overexpression is not a substantial factor in the observed flux measurements.

Fig. 2. Comparison of metabolite ratios between strains grown under regular growth conditions (left panels) and strains propagated in the presence of labeled glucose (right panels). (A) 2-oxoglutarate/succinate. (B) 2-oxoglutarate/fumarate. (C) 2-oxoglutarate/malate. (D) 2-oxoglutarate/oxaloacetate. (E) 2-oxoglutarate/glutamine. (F) pyruvate/oxaloacetate. Asterisks indicate significant differences ($p < 0.05$, two-tailed Mann-Whitney test). ns, non-significant. Note that the ratios in strains carrying the empty vector and the strains expressing orthologous MDH proteins are identical. BsMDH/inactBsMDH, active/inactive *B. subtilis* MDH; StMDH/inactStMDH, active/inactive *S. sciuri* MDH; OceMDH/inactOceMDH, active/inactive *O. iheyensis* MDH; control, *B. subtilis* carrying empty pHT254 vector.

5. *The conclusion only summarizes the findings. I want to know more about the impact of the findings, still missing information, and future directions.*

As requested, we have expanded the Conclusion section, and added the potential impact of the findings and future directions (lines 496-504 in the revised manuscript).

Reviewer #2

*In this manuscript, Jasinska et al. study the relevance of protein-protein interactions between enzymes that catalyze non-consecutive steps in a metabolic pathway. It is generally assumed that enzymes of metabolic pathways assemble to so-called metabolons. Typically, these interactions are weak and difficult to reconstitute in vitro. It is also assumed that the interaction between enzymes that catalyze consecutive reactions of the pathway serves to shuttle metabolic intermediates directly from one enzyme to the next. However, the published interaction between the malate dehydrogenase (MDH) and isocitrate dehydrogenase (ICD) in the TCA cycle in the bacterium *B. subtilis* doesn't meet this criterium as the enzymes catalyze distant steps in the pathway thus questioning the relevance of the interaction. A previous study by Bartholomae et al. (2014) has already shown that the complex formation results in enhanced ICD activity. This result is confirmed in the current study using a different experimental approach as compared to the original paper. Using a combination of experimental and mathematical analysis the authors show that the interaction results in ICD clustering which is responsible for the enhanced enzyme activity. In addition, the authors show that overexpression of MDH results in accumulation of 2-oxoglutarate which is unexpectedly accompanied by reduced fluxes in associated reactions (such as amino acid transamination). Unfortunately, the reason for the latter observation could not be clarified.*

In the paper by Bartholomae et al. (2014) a mutant version of ICD was used under diluted conditions and no mechanistic explanation for the observation was offered. Here we provide a clear mechanism, supported by experiment and theory, for the rate enhancement by ICD under conditions resembling crowded intracellular environment. Furthermore, we predict and observe in vivo a drop in flux downstream to 2-oxoglutarate junction due to accumulation of the ICD reaction product in the vicinity of MDH-ICD clusters.

Reviewer #3

1. Lines 92-99: It is intriguing that MDH and ICD can interact... but in vivo, it is likely that other enzymes catalyzing intermediary steps (e.g. CS and CAN, mentioned on line 245 for CS) also form part of the supramolecular assembly. This should be mentioned here, as direct substrate channeling may well represent one of the metabolon properties when considering the whole picture.

We agree. We added this notion to the main text (lines 99-101 in the revised manuscript).

Line 122: The claim that “liquid-liquid phase separated protein droplets” resemble the intracellular environment seems a bit bold to me, or at least it needs some justification (Is it because of the high molecular crowding? Of other biochemical properties? Re-creating the cellular environment in vitro represents a long-lasting challenge, see for example: <https://doi.org/10.1016/j.pisc.2014.02.011>).

We fully agree with the reviewer that a more detailed explanation as to why LLPS system resembles the intracellular environment is needed. In the revised version of the manuscript (125-129), we elaborate on the similarities in crowding, viscosity, and pH values between our droplet system and the cellular environment. The protein concentration within liquid-liquid phase-separated droplets closely resembles the protein *crowding* concentrations found within cells. In detail, the concentration of proteins within the droplets is around 4.8 ± 0.5 mM (320 ± 10 mg/mL), which is similar to that reported in the cytoplasm⁴⁻⁷. The *viscosity* of the droplet phase determined by particle tracking micro-rheology was found to be 2.1 Pa⁸, which is about 2000-fold higher than that of water, and is comparable to values reported for the cytoplasm^{9, 10}. Lastly, the *pH* within protein droplets was previously measured using the pH sensitive probe SNARF at the confocal microscope and was found to span between values of 7.1 and 7.4⁸. These pH values closely resemble the intracellular pH of most organisms^{11, 12}. In conclusion, the physicochemical parameters of the LLPS system closely resemble those found within cells.

Figure 1 (and corresponding text): The percentage of rate enhancement when adding B. subtilis MDH saturates at high MDH:ICD ratios (somewhere around 200). Are such high ratios really physiologically relevant? What happens when much lower ratios (e.g. 1:1 or 5:1 ... or even 1:5, any data available?) are utilized, which probably mimics better what would happen in the intracellular environment?

We thank the reviewer for the comment. To address the question, we performed new experiments in the droplets using lower MDH:ICD ratios. In detail, we used 1:1 and 10:1 MDH/ICD molar ratios. Whereas no changes can be observed at a 1:1 ratio, at a 10:1 MDH/ICD ratio, which we also used in our *in vivo* experiment, we observed a ~10% rate enhancement as well as a measurable product accumulation (Fig. 3 below). We have added the new data to panels B and C of Fig. 1 in the main text. In panel B we have added the two points obtained by using the MDH:ICD ratios of 1:1 and 10:1, while in panel C we have implemented in the image the accumulation of 2-oxoglutarate measured within the droplets.

Fig. 3. (Fig. 1 in the main text) Interaction between *B. subtilis* ICD and MDH within protein droplets causes ICD rate enhancement and product partitioning. (A) Confocal microscopy images of ICD and MDH showing the co-localization of the enzymes within droplets. ICD and MDH were labeled with Alexa Fluor 594 (red) and Alexa Fluor 488 (green), respectively. (B) Rate of 2-oxoglutarate formation (expressed as % of activity) of ICD within droplets is increased in the presence of increasing amounts of *B. subtilis* MDH (1:1, 10:1, 50:1, 100:1, 250:1, and 500:1 MDH/ICD molar ratios), while no effects have been observed in the presence of orthologous MDHs from *O. iheyensis* and *S. sciuri* at the same MDH/ICD ratios. Inset: rate of 2-oxoglutarate formation expressed as ICD velocity (sec^{-1}) in the presence of *B. subtilis* and orthologous MDHs. (C) An increase in MDH/ICD ratios within droplets (1:1, 10:1, 50:1, 100:1, 250:1, and 500:1 MDH/ICD molar ratios) causes an increase in delay of 2-oxoglutarate diffusion into the continuous phase, suggesting that 2-oxoglutarate is sequestered within droplets. (D) Cartoon of LLPS system that models the reported results. The ICD and MDH enzymes partition within the BSA droplets and the small molecules (substrate, S and product, P) freely diffuse in and out. In the presence of increasing amounts of *B. subtilis* MDH (BsMDH), ICD catalytic rate increases, probably due to MDH-induced clustering of ICD molecules. An increase in ICD activity within clusters causes local accumulation of the reaction product, manifested in the apparent partitioning of 2-oxoglutarate between the droplet and continuous phases.

Panel 1D, right: I got a bit confused because it looks like the quantity of ICD within the droplet increases after MDH addition (schematically, 5 ICDs on the left but 7 on the right). However, the quantity of ICD is not supposed to change, right? This panel should be modified.

We thank the reviewer for noticing the discrepancy with the number of ICD molecules. We have now modified the schema of panel D with an equal number of ICD molecules in Fig. 1 of the revised manuscript (Please see also Fig. 3 above).

Line 138: Could you also indicate the percentage of identity between ICDs from these different organisms? Is there any information available as to whether the MDH-ICD interactions also exist within these organisms?

This is an interesting question. The identity in amino acid sequence between ICD orthologs in *B. subtilis* and *S. sciuri* pair and between *B. subtilis* and *O. iheyensis* pair is rather high and constitutes 83%. We introduced this information in the main text (lines 148-150 in the revised version). No information about MDH-ICD cluster formations in these organisms have been reported. We are planning to investigate this question in the future using our LLPS experimental setup.

Lines 158-160 and Fig.S3: Signal intensity within the droplet diminishes in presence of *B. subtilis* MDH and it seems that some “fluorescent puncta” concomitantly appear (Fig.S3B lower panel). Based on confocal microscopy observations, are these puncta indeed only visible in presence of *B. subtilis* MDH? Are these puncta also visible at lower MDH:ICD ratios? This is important because it could represent the only direct experimental evidence of ICD clustering...

We thank the reviewer for the comment. We have reported in Fig. S3 in Supplementary Materials that only in the presence of *B. subtilis* MDH the “fluorescent puncta” appear, while adding MDHs from orthologues does not produce any visible fluorescent puncta are visible (Please see right side of panel B in Fig. 4 below or in Fig. S3 in Supplementary Materials). Along with this analysis, the signal of the fluorescence in the presence of orthologues MDHs shows no changes, while in the presence of *B. subtilis* MDH the signal decreases within the droplets due to fluorescent puncta formation.

Fig. 4. (Fig. S3 in Supplementary Materials). Qualitative analysis of ICD clustering mediated by MDH. (A) Confocal images and calibration curve of the signal intensity mediated by Alexa Fluor 594 labeled ICD at three different concentrations (0.1 μM, 0.5 μM and 1 μM). (B) Confocal images of ICD and analysis of the fluorescence intensity inside phase-separated droplets containing only *B. subtilis* ICD or coupled with a large excess of the three different MDHs employed in this study. ICD concentration within the droplets is around 2-2.5 μM while MDH concentration is around 400-500 μM. At least 5-10 droplets for each condition were analyzed.

Also, as suggested by the reviewer, we have prepared and analyzed the confocal microscope images by using lower MDH:ICD ratios (1:1, 10:1). We found that at these ratios no changes associated with fluorescence intensity or puncta formation are observed (Fig. 5 below). We added the new experiments (panel C) to Fig. S3 in the Supplementary Material.

Fig. 5. Confocal images of ICD and analysis of the fluorescence intensity inside phase-separated droplets containing only *B. subtilis* ICD or coupled with a large excess of the three different MDHs employed in this study. ICD concentration within the droplets was around 2-2.5 μM while MDHs have been used for the ratio 1:1 at 2.5 μM , ratio 10:1 at 25 μM and ratio 250:1 at 500 μM . Scale bar is 20 μm . At least 5-10 droplets for each condition have been analyzed.

Line 213-215: If I understand correctly, the production rate within the cluster is higher than the diffusion rate of 2-oxoglutarate outside the cluster, explaining why the local concentration increases? This should be stated more clearly.

This is exactly right. We clarified this point in the main text accordingly (lines 222-224 in the revised text).

Figure 2: Is Fig. 2D showing actual experimental data? Please add “measured product concentration” so the reader can directly see these are empirical results, not results from modelling. Same thing about panels 2D and 2F (I assume these are experimental results but it is not very clear based on the legend... also, what do the error bars represent?).

We thank the reviewer for noticing the discrepancy. The reason for confusion is the mislabeling in the figure caption. We corrected the mistakes and updated the figure (see Fig. 6 below and Fig. 2 in the main text of revised manuscript). We also added the meaning of error bars, which represent STD of three independent measurements.

Fig. 6. (Fig. 2 in the revised main text). Enzyme clustering leads to an increase in product concentration within clusters. (A) Spherically symmetric model of enzyme concentration, whereby enzyme E form clusters of radius R_1 within droplets of radius R_2 . The droplet is surrounded by bulk with radius R_0 . (B) Predicted distribution of product with the different regions of the droplet and basin. The calculations are shown for $\gamma = 10$, $\gamma = n_1/n_2$ (10-fold more enzyme E molecules are found within the cluster than in the droplet outside the cluster) (C) *Left panel*. Predicted distribution ratio of total product concentration within droplets vs. bulk as a function of the enrichment of enzyme E within the cluster. Ratios in x axes correspond to the molar excess of MDH relative to ICD. *Right panel*. Experimental increase in product concentration in droplets vs. bulk as a function of increasing ICD cluster formation mediated by MDH. Error bars represent STD of three independent measurements. Note the agreement between theory and experiment. (D) *Left panel*. Predicted concentration of a product in bulk and in droplets as a function of increasing cluster formation. Ratios in x axes correspond to the molar excess of MDH relative to ICD. *Right panel*. Experimental increase in product concentration in bulk and droplets. Error bars represent STD of three independent measurements. Note the agreement between theory and experiment. (E, F) Modeling of cluster formation in vivo. (E) Formation of ICD cluster within the cytoplasm (cluster radius is assumed to be $R_1 = 0.1 \mu\text{m}$, cytoplasm size is $R_2 = 2 \mu\text{m}$, see also Supporting Information Table S3) causes an increase in product concentration within and in the immediate proximity of the cluster. The calculations are shown for $\gamma = 100$ (most of the ICD molecules are found within the cluster). Zero microns correspond to the cluster's center. Input corresponds to the reaction substrate. (F) Predicted substrate concentration (C1) within the cluster and the cytoplasm as a function γ .

Lines 224-225: You hypothesize that substrate is present in a large excess. Is it really compatible with what happens intracellularly, i.e. where only limited amounts of isocitrate are available? Please add a sentence to discuss about the last point, because it is important concerning the in vivo relevance of the proposed model.

The limited amounts of isocitrate in vivo are most probably a result of high consumption rate, rather than low production rate. We assume that under the excess amounts of carbon source (glucose) in the growth medium used in our experiments, the production rate of citrate is high enough not to constitute a limiting factor within clusters. As requested, we added this notion to the text (lines 241-242 in the revised manuscript).

Line 240: Can you really state based on your model that “clustering should cause rate enhancement of ICD activity”? Clustering does not seem to have any effect on the ICD (specific) activity, it rather leads to a local increase in ICD concentration and therefore in 2-oxoglutarate formation. I would be careful with phrasing.

The rate enhancement is indeed caused by the local increase in ICD concentration within clusters, and not by an intrinsic change in ICD properties (such as specific activity). Following the reviewer’s concern, we changed the term ‘rate enhancement’ to an ‘apparent rate enhancement.’

Fig. S5: Panels A) and B) seem to be inverted in the figure.

Thank you for noticing. We corrected the mistake.

Fig.3E and comparison with Fig. S9: If I understand correctly, these heat maps refer to results of targeted metabolomic experiments performed on separate occasions. There seems to be some discrepancies within the results when looking into details: in Fig.3E 2-oxoglutarate (and pyruvate) appear relatively more abundant in the BsMDH (active or inactive) expressing strains than in the controls, but the tendency is not clear in Fig.S9. How reproducible are these metabolomic experiments (besides the PCA showing a comparable separation of samples) and can we really conclude that there is a 2-oxoglutarate accumulation when over-expressing BsMDH?

The technical noise produced by MS measurements is a known problem in metabolomics. Therefore, a direct comparison of individual metabolite intensities between replicate measurements may show a variability, as indeed is seen between datasets in Fig. 3 (main text) and Fig. S9 (Supplementary Materials). A widely used approach to reduce the impact of the instrumental noise (apart from the PCA analysis) is to compare **ratios** between metabolite intensities within each dataset rather than intensities themselves¹³. This approach is based on the assumption that if the overall signal intensity changes between samples injected in MS, the ratios between metabolites within the sample should be robust to such a change. We followed this approach and compared ratios of metabolites across the two datasets. We focused, in

particular, on metabolites directly or indirectly related to fluxes flowing through the 2-oxoglutarate junction. The results are shown in Fig. 7 below (and in Fig. S10 in the revised Supplementary Materials). A clear conservation of ratios between the two datasets supports the validity of these measurements. We added the description of the analysis to the revised main text (lines 316-317)

Fig. 7. (Fig. 10 in the revised Supplementary Materials). Comparison of changes in metabolite ratios between two replicate experiments. (A-I) Left panel. Data are related to the experimental dataset presented in Fig. 3E (main text). *Right panel.* Data are related to the experimental dataset presented in Fig. S9 (Supplementary

Materials). Note the similarity in the ratio patterns between the two experiments. Asterisks indicate significant differences ($p < 0.05$, two-tailed Mann Whitney test). ns, non-significant. BsMDH/inactBsMDH, active/inactive *B. subtilis* MDH; StMDH/inactStMDH, active/inactive *S. sciuri* MDH; OceMDH/inactOceMDH, active/inactive *O. iheyensis* MDH; control, *B. subtilis* carrying empty pHT254 vector.

Lines 393-394: Overexpressing BsMDH may have other pleiotropic effects that could only be assessed using a systems biology approach (e.g. transcriptomics, quantitative proteomics...). It is possible that the highly accumulated BsMDH triggers regulatory responses and that the abundance/activity of other key central metabolism enzymes consequently changes. The impacts would go far beyond a “stoichiometric disruption of the TCA cycle supramolecular complex” and could also explain the observations made in presence of deimethyl-2-oxoglutarate.

To validate that the observed metabolic changes are not caused by unrelated pleiotropic effects triggered by BsMDH overexpression, we decided to perform an additional control experiment. To this end, glucose, the main carbon source in the growth medium used in MDH overexpression experiments, was replaced with glutamine. In *B. subtilis* glutamine is converted directly to glutamate through the activity of glutaminase^{14,15}, thus potentially alleviating the 2-oxoglutarate limitation triggered by MDH overexpression. We reasoned, therefore, that propagation on glutamine should not be accompanied by the metabolic imbalance observed with glucose. If true, we could exclude the pleiotropic effects of MDH overexpression as the main culprit in our observations. After validating that the levels of MDH overexpression are maintained upon glucose replacement with glutamine (see Fig. 8 below and Fig. S12 in Supplementary Materials), we propagated the strains carrying empty vector, endogenous and orthologous MDHs in the presence of glutamine and conducted targeted and untargeted metabolomic analyses. We found that replacement of glucose with glutamine fully removed the metabolic imbalance observed in the presence of glucose, as can be seen in the individual metabolite levels (see Fig. 9 below and Fig. S13 in Supplementary Materials), heat maps and PCA analysis (see Fig. 10 below and Fig. S14 in Supplementary Materials), and metabolite ratios (see Fig. 11 below and Fig. S15 in Supplementary Materials). We, therefore, conclude that the phenomena observed upon BsMDH overexpression is unlikely to be caused by pleiotropic effects. We added the description of the experiment and its conclusion to the main text (lines 347-363), Materials and Methods, and Supplementary figures of the revised manuscript.

Fig. 8. (Fig. S12 in revised Supplementary Materials). Western blot analysis of MDH expression levels in lysates of cultures propagated on glutamine as the main carbon source. 1,2 - active, inactive *S. sciuri* MDH; 2,3 – active, inactive *O. iheyensis* MDH; 3,4 – active, inactive *B. subtilis* MDH. The detection was done with anti-His antibodies.

Fig. 9. (Fig. S13 in revised Supplementary Materials). (A) PCA of untargeted metabolite intensities identified in *B. subtilis* replicate populations (four of each kind) expressing either active or inactive endogenous and orthologous MDH proteins and control populations carrying empty pHT254 vector. The cultures were propagated on glutamine as the main carbon source and induced with IPTG. (B) Heat map of $-\log_2$ of relative intensities of untargeted metabolites. BsMDH/inactBsMDH, active/inactive *B. subtilis* MDH; StMDH/inactStMDH, active/inactive *S. sciuri* MDH; OceMDH/inactOceMDH, active/inactive *O. iheyensis* MDH; control, *B. subtilis* carrying empty pHT254 vector.

Fig. 10. (Fig. S14 in revised Supplementary Materials). Relative intensities of the identified metabolites upon MDH overexpression with glutamine as the main carbon source. (A-H) $-\log_{10}$ of intensities of individual metabolites. No significant changes between strains can be found (two-tailed Mann Whitney test). BsMDH/inactBsMDH, active/inactive *B. subtilis* MDH; StMDH/inactStMDH, active/inactive *S. sciuri* MDH; OceMDH/inactOceMDH, active/inactive *O. iheyensis* MDH; control, *B. subtilis* carrying empty pHT254 vector. BsMDH/inactBsMDH, active/inactive *B. subtilis* MDH; control, *B. subtilis* carrying empty pHT254 vector.

Fig. 11. (Fig. S15 in revised Supplementary Materials). Ratios of relative intensities of targeted metabolites in propagated on glutamine as the main carbon source. (A-I) BsMDH/inactBsMDH, active/inactive *B. subtilis* MDH; StMDH/inactStMDH, active/inactive *S. sciuri* MDH; OceMDH/inactOceMDH, active/inactive *O. iheyensis* MDH; control, *B. subtilis* carrying empty pHT254 vector. No significant differences in metabolite ratios between strains can be detected (two-tailed Mann Whitney test).

References

1. Guiziou, S. et al. A part toolbox to tune genetic expression in *Bacillus subtilis*. *Nucleic Acids Res* **44**, 7495-7508 (2016).
2. Phan, T.T., Tran, L.T., Schumann, W. & Nguyen, H.D. Development of Pgrac100-based expression vectors allowing high protein production levels in *Bacillus subtilis* and relatively low basal expression in *Escherichia coli*. *Microb Cell Fact* **14**, 72 (2015).
3. Meyer, F.M. et al. Physical interactions between tricarboxylic acid cycle enzymes in *Bacillus subtilis*: evidence for a metabolon. *Metab Eng* **13**, 18-27 (2011).
4. Ellis, R.J. Macromolecular crowding: obvious but underappreciated. *Trends Biochem Sci* **26**, 597-604 (2001).
5. Goodsell, D.S. Inside a living cell. *Trends Biochem Sci* **16**, 203-206 (1991).
6. Milo, R. What is the total number of protein molecules per cell volume? A call to rethink some published values. *Bioessays* **35**, 1050-1055 (2013).
7. Milo, R. & Phillips, R. Cell biology by the numbers. (Garland Science, Taylor & Francis Group, New York, NY; 2016).
8. Testa, A. et al. Sustained enzymatic activity and flow in crowded protein droplets. *Nat Commun* **12**, 6293 (2021).
9. Guo, M. et al. Probing the stochastic, motor-driven properties of the cytoplasm using force spectrum microscopy. *Cell* **158**, 822-832 (2014).
10. Nishizawa, K. et al. Universal glass-forming behavior of in vitro and living cytoplasm. *Sci Rep* **7**, 15143 (2017).
11. Krulwich, T.A., Sachs, G. & Padan, E. Molecular aspects of bacterial pH sensing and homeostasis. *Nat Rev Microbiol* **9**, 330-343 (2011).
12. Slonczewski, J.L., Fujisawa, M., Dopson, M. & Krulwich, T.A. Cytoplasmic pH measurement and homeostasis in bacteria and archaea. *Adv Microb Physiol* **55**, 1-79, 317 (2009).
13. Weckwerth, W., Loureiro, M.E., Wenzel, K. & Fiehn, O. Differential metabolic networks unravel the effects of silent plant phenotypes. *Proc Natl Acad Sci U S A* **101**, 7809-7814 (2004).
14. Brown, G. et al. Functional and structural characterization of four glutaminases from *Escherichia coli* and *Bacillus subtilis*. *Biochemistry* **47**, 5724-5735 (2008).
15. Satomura, T. et al. Enhancement of glutamine utilization in *Bacillus subtilis* through the GlnK-GlnL two-component regulatory system. *J Bacteriol* **187**, 4813-4821 (2005).

Reviewers' Comments:

Reviewer #1:

Remarks to the Author:

I find all my concerns are addressed in the authors' responses. However, I suggest adding these explanations to the manuscript to avoid the doubt like those I had. The protein expression levels and their effects on cellular metabolisms should be clearly mentioned in the manuscript.

Following are the comments for the responses to my major concerns in the first round.

1. Thank you for the precise explanation. I agree with the use of the selected promoter. I suggest including some sentences to explain the nature of the promoter and add Fig 1 as a supplementary figure.
2. I understand the difficulties of showing LLPS in vivo. Please clarify that LLPS including MDH/ICD is not experimentally shown and is a future research topic.
3. 4. As the MDH expression is moderate, I agree with the argument.
5. Thank you for sharing your perspectives.

Reviewer #3:

Remarks to the Author:

All my previous comments have been addressed, with novel experimental results added when appropriate. The study is well-written and interesting, highlighting the importance of MDH-ICD interactions in metabolon function. Nice work!

A couple of very minor corrections to do:

- Legend of Figure 2: for both (C) and (D), the sentence "Ratios in x axes correspond to the molar excess of MDH relative to ICD." should be put in the sections corresponding to the right panels, not the left ones.
- Lines 243-244: I found this sentence a bit weird after re-reading it. If $\gamma = 100$, by definition most of the ICD molecules are in the cluster, this is not surprising.

We thank the editors and the reviewers for the second revision of our manuscript. Our point-by-point responses to the reviewers' comments are below; their original comments are in italics. All changes introduced to the revised text are highlighted in red.

Reviewer #1

1. Thank you for the precise explanation. I agree with the use of the selected promoter. I suggest including some sentences to explain the nature of the promoter and add Fig 1 as a supplementary figure.

As requested, we added the explanation to the main text (lines 275-276) and added Fig. 1 as a Supplementary figure (Supplementary Fig. 5B in the revised version).

2. I understand the difficulties of showing LLPS in vivo. Please clarify that LLPS including MDH/ICD is not experimentally shown and is a future research topic.

We added a specific notion regarding the difficulties of showing LLPS in vivo and mentioned that this is a topic of future research (lines 500-501).

3. 4. As the MDH expression is moderate, I agree with the argument.

5. Thank you for sharing your perspectives.

Reviewer #3

- Legend of Figure 2: for both (C) and (D), the sentence "Ratios in x axes correspond to the molar excess of MDH relative to ICD." should be put in the sections corresponding to the right panels, not the left ones.

We corrected the mistake (lines 948, 953 in the revised version)

- Lines 243-244: I found this sentence a bit weird after re-reading it. If $\gamma = 100$, by definition most of the ICD molecules are in the cluster, this is not surprising.

We agree that this is a trivial sentence; we removed it entirely from the text.